# Increased glucose availability sensitizes pancreatic cancer to chemotherapy

Ali Vaziri-Gohar [1,2] ✉, Jonathan J. Hue[3], Ata Abbas [1], Hallie J. Graor[1], Omid Hajihassani[1], Mehrdad Zarei [1,3], George Titomihelakis[4], John Feczko[5], Moeez Rathore[1], Sylwia Chelstowska[1], Alexander W. Loftus[3], Rui Wang[1], Mahsa Zarei[6], Maryam Goudarzi [7], Renliang Zhang[7], Belinda Willard[7], Li Zhang[8], Adam Kresak[1,9], Joseph E. Willis[9], Gi-Ming Wang[10], Curtis Tatsuoka[11], Joseph M. Salvino [12], Ilya Bederman[5], Henri Brunengraber[13], Costas A. Lyssiotis[8], Jonathan R. Brody[14] & Jordan M. Winter [1,3] ✉

Pancreatic Ductal Adenocarcinoma (PDAC) is highly resistant to chemotherapy. Effective alternative therapies have yet to emerge, as chemotherapy remains the best available systemic treatment. However, the discovery of safe and available adjuncts to enhance chemotherapeutic efficacy can still improve survival outcomes. We show that a hyperglycemic state substantially enhances the efficacy of conventional single- and multi-agent chemotherapy regimens against PDAC. Molecular analyses of tumors exposed to high glucose levels reveal that the expression of GCLC (glutamate-cysteine ligase catalytic subunit), a key component of glutathione biosynthesis, is diminished, which in turn augments oxidative anti-tumor damage by chemotherapy. Inhibition of GCLC phenocopies the suppressive effect of forced hyperglycemia in mouse models of PDAC, while rescuing this pathway mitigates anti-tumor effects observed with chemotherapy and high glucose.

Approved multi-agent chemotherapy regimens offer a marginal advance over single-agent treatments, and resistance to these cocktails virtually always develops[1,2]. Specifically, FOLFIRINOX (FFX; folinic acid, 5-FU (5-fluorouracil), irinotecan, and oxaliplatin) or a doublet of gemcitabine plus albumin-bound (nab) paclitaxel confers a median survival benefit of roughly 4-month over single-agent gemcitabine[3,4]. The overall median survival of patients with metastatic pancreatic cancer in the modern era is just 8–11 months, and the five-year survival rate is around 3%[5]. The relative ineffectiveness of standard chemotherapy (standard of care; SOC) in pre-clinical murine PDAC models also mirrors these poor patient outcomes[6,7].

In light of the fact that targeted agents and immunotherapeutics have failed to positively impact patients with PDAC, chemotherapy, therefore, remains the mainstay of treatment. Strategies designed to

[1]Case Comprehensive Cancer Center, Case Western Reserve University, Cleveland, OH, USA. [2]Department of Cancer Biology, Cardinal Bernardin Cancer Center, Stritch School of Medicine, Loyola University Chicago, Maywood, IL, USA. [3]Department of Surgery, Division of Surgical Oncology, University Hospitals, Cleveland Medical Center, Cleveland, OH, USA. [4]Jefferson Pancreas, Biliary and Related Cancer Center, Thomas Jefferson University, Philadelphia, PA, USA. [5]Department of Genetics and Genome Sciences, Case Western Reserve University, Cleveland, OH, USA. [6]Department of Veterinary Physiology and Pharmacology, Texas A&M University, College Station, TX, USA. [7]Proteomics and Metabolomics Core, Cleveland Clinic, Cleveland, OH, USA. [8]Department of Molecular and Integrative Physiology, University of Michigan School of Medicine, Ann Arbor, MI, USA. [9]Department of Pathology, Case Western Reserve University and Department of Pathology Cleveland Medical Center, Cleveland, OH, USA. [10]Department of Population and Quantitative Health Sciences, Case Western Reserve University, Cleveland, OH, USA. [11]Department of Medicine, Division of Hematology/Oncology, University of Pittsburgh, Pittsburgh, PA, USA. [12]Molecular and Cellular Oncogenesis Program, The Wistar Institute, Philadelphia, PA, USA. [13]Department of Nutrition and Biochemistry, Case Western Reserve University, Cleveland, OH, USA. [14]Brenden Colson Center for Pancreatic Care; Departments of Surgery and Cell, Developmental & Cancer Biology; Knight Cancer Institute, Oregon Health and Science University, Portland, OR, USA. ✉e-mail: avaziri1@luc.edu; jordan.winter@UHhospitals.org

increase the potency of available chemotherapy agents are an attractive, yet relatively understudied investigative approach[8]. For instance, a consortium of leading PDAC foundations in North America is co-sponsoring a phase II randomized trial (PASS-01) to identify predictive markers that select patients for the optimal chemotherapeutic regimen (NCT04469556). The identification of readily available, safe, and effective adjuncts to enhance the activity of existing chemotherapies may also improve patient survival, and perhaps avoid the financial, technical, and regulatory challenges associated with the development of new compounds.

One possible reason for the poor anti-tumor activity of chemotherapy is that pancreatic tumors are extremely desmoplastic[9]. This pathologic feature is associated with low microvascular density[6,10], tissue hypoxia[11], and steep nutrient gradients[12–14]. To thrive under such harsh conditions, cancer cells require specific molecular adaptations, including enhanced utilization of alternative energy substrates[15–18], optimized mitochondrial function[14,19], and improved handling of reactive oxygen species[14,20]. Our group previously demonstrated that an RNA-binding protein, HuR (ELAVL1), translocates from the nucleus to the cytoplasm under acute metabolic stress, such as glucose or glutamine withdrawal. In this context, HuR supports key survival pathways[21]. Additionally, these adaptive mechanisms lead to acquired chemoresistance[21,22]. Due to the acute stress response, including the HuR pro-survival network, chemotherapy actually becomes less effective under nutrient-deprived, PDAC-associated conditions. Chemoresistance under nutrient withdrawal is reproducibly apparent in cell culture PDAC models[21]. Consistent with these observations, patients receiving chemotherapy experienced worse survival after surgery when their peripheral glucose levels were in the normal range (i.e., which translates to an especially nutrient-deprived microenvironment and chemoresistance), as compared to hyperglycemic patients, which harbor tumors with relatively higher levels of ambient glucose[21]. These observations collectively suggest that serum glucose levels are associated with chemotherapy response. For example, elevated serum glucose predicts chemosensitivity. This invites an intriguing question: can intentional or forced hyperglycemia be leveraged to sensitize PDAC to chemotherapy? Herein, we test this hypothesis, and further, attempt to elucidate the molecular underpinnings of hyperglycemia-associated chemo-sensitization in order to expose therapeutic targets.

## Results

### Increased survival rate in PDAC patients with diabetes compared to patients without diabetes

We first validated a prior retrospective clinical study by our group performed on a cohort of patients with localized PDAC[21]. In the present series, we examined the impact of glycemic status on an independent group of patients with metastatic PDAC. Approximately 33% of patients in a single-institution experience presented with elevated glucose levels (at least one glucose reading above 200 mg/dL), consistent with historical populations[23]. There were no appreciated demographic differences between normal and high glucose patients (Supplementary Table 1). A greater proportion of patients in the high glucose group carried an established or documented clinical diagnosis of diabetes mellitus at the time of PDAC diagnosis, as compared to the normal glucose group (56.2% vs. 14.8%, $P < 0.001$). Median glucose levels pre-diagnosis (137 vs. 105 mg/dL, $P < 0.001$) and during treatment (158 vs. 109 mg/dL, $P < 0.001$) were higher among patients in the high glucose group. The median CA 19-9 level, a prognostic marker commonly used to reflect disease burden and PDAC aggressiveness, was similar between groups at diagnosis, although it actually trended towards a higher value in the high glucose group (2439.5 U/mL high glucose vs 1294.5 U/mL normal glucose, $P = 0.224$). First-line chemotherapy regimens were similar as well, with the majority of patients in the entire cohort receiving multi-agent standard-of-care regimens ($P = 0.881$). The median overall survival among all patients who completed at least two cycles of chemotherapy was approximately 9.8 months in the overall cohort (IQR: 6.3, 14.9 months), which is on par with historical clinical trial data[3,4]. Multivariable Cox proportional hazards regression demonstrated that patients in the high glucose level cohort had a relatively greater survival, despite a higher CA19-9 level, as compared to patients in the normal glucose group (HR = 0.61, 95% CI 0.41–0.92, $P = 0.02$) (Fig. 1a). Notably, no associated survival difference was observed based on glucose levels in an independent cohort of metastatic patients who did not receive treatment (HR = 0.99, 95% CI 0.64-1.53, $P = 0.97$) (Fig. 1b), suggesting that the interaction with glycemic status may be present only for patients who receive chemotherapy. Medical comorbidities, performance status, site of metastatic disease, CA 19-9 value at diagnosis, the total number of chemotherapy cycles, and chemotherapy regimen are provided, and do not reveal any obvious confounders (Supplementary Table 2). Paired with prior cell culture and clinical data[21], these data indicate that a high-glucose state sensitizes PDAC to conventional chemotherapy.

### Hyperglycemia augments chemo-efficacy in pre-clinical PDAC models

We followed up on these findings with a series of controlled studies in various mouse PDAC models of hyperglycemia. First, we induced hyperglycemia pharmacologically using streptozotocin (STZ)[24]. This drug chemically ablates β-islet cells in the pancreas to induce pancreatogenic diabetes. Since this model results in extremely elevated

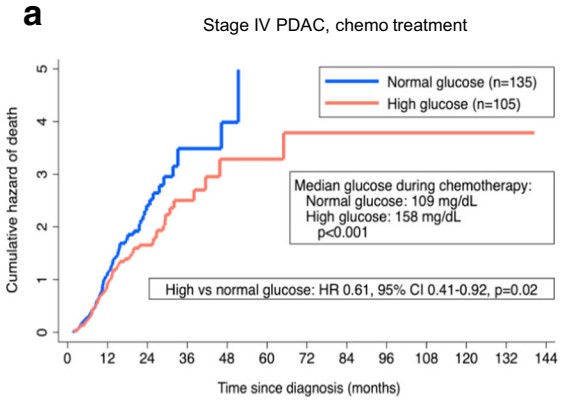
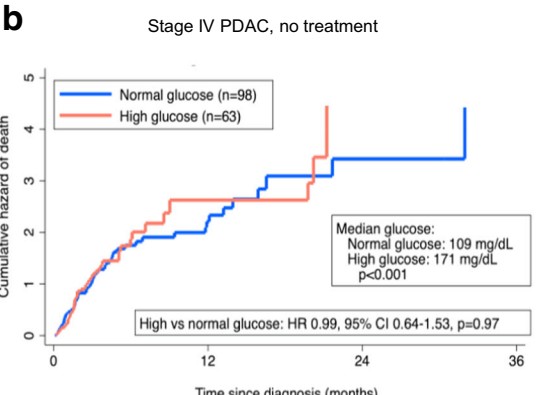

**Fig. 1 | Enhanced chemotherapy response with hyperglycemia in patients with stage IV PDAC.** Nelson-Aalen cumulative hazard curves of patients with metastatic PDAC, stratified by glycemic status among patients who received ≥ 2 cycles of chemotherapy (a) and those who did not receive any treatment (b). Patients' data are provided in the Source Data file.

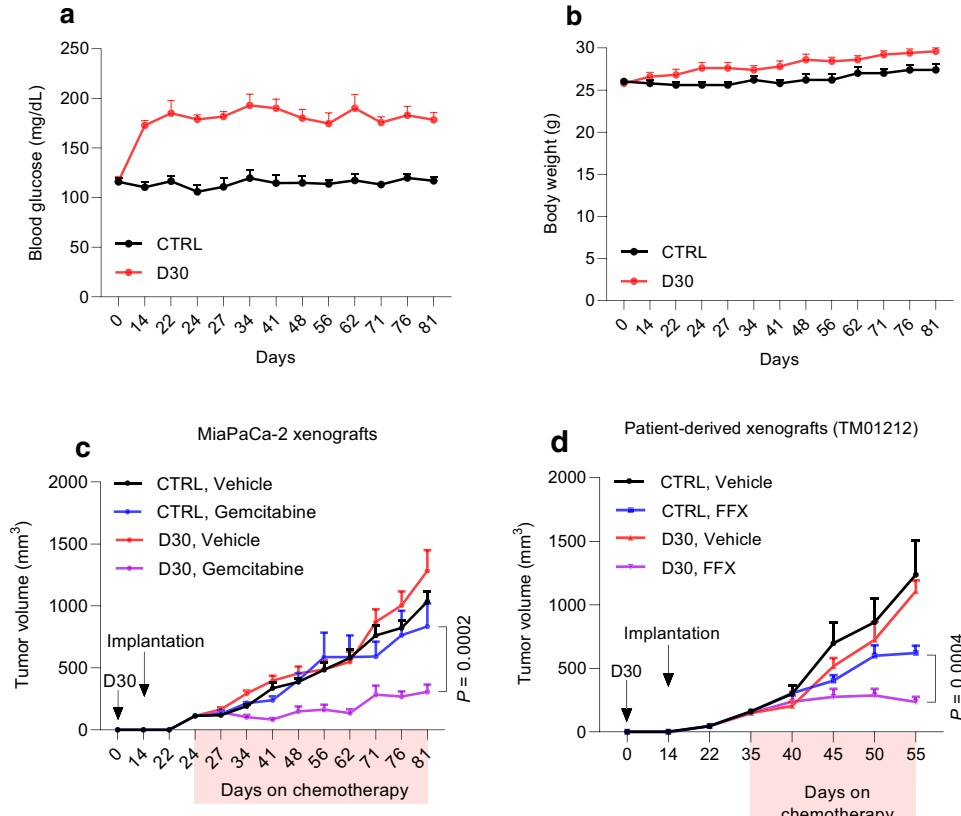

**Fig. 2 | High-glucose diet in mice improves conventional chemotherapy response. a, b,** Peripheral glucose levels (**a**) and body weights (**b**) of nude mice receiving normal or D30 water. Each dot represents the mean of weekly measurements of blood glucose per group ($n = 5$ mice per group). **c**, Xenograft growth of MiaPaCa-2 cells treated with gemcitabine ($n = 5$ mice per group, 75 mg/ kg twice weekly, i.p.). **d,** Patient-derived xenograft (TM01212) treated with FOL-FIRINOX ($n = 5$ mice per group, oxaliplatin 5 mg/kg, 5-FU 25 mg/kg, and irinotecan 50 mg/kg, once weekly, i.p.). Data are provided as mean ± s.e.m. Longitudinal mixed models were fit for tumor size growth, and time by treatment interactions were assessed (**c, d**). Source data are provided as a Source Data file.

and debilitating glucose levels, titration to non-toxic levels of hyperglycemia was necessary with daily injections of long-acting insulin glargine (Supplementary Fig. 1a). In separate experiments, hyperglycemia was also induced in independent experimental arms through dietary changes. Specifically, mice were allowed to consume 30% dextrose water (D30) *ad libitum*. We previously validated this hyperglycemic model in a flank xenograft model through serial peripheral glucose monitoring, along with elevated intra-tumoral glucose measurements[14]. Peripheral glucose levels and mouse weights associated with each hyperglycemic model are provided here for validation (Fig. 2a, b, and Supplementary Fig. 1b, c), and demonstrate increased peripheral glucose levels with stable body weights, as seen in our prior studies[14]. PDAC xenografts in hyperglycemic nude mice exhibited greater sensitivity to single-agent chemotherapy in the two independent models of forced hyperglycemia (Fig. 2c and Supplementary Fig. 1d, 2). As observed previously with patients in the absence of chemotherapy (Fig. 1b), no significant differences in growth rates were observed with forced hyperglycemia in the vehicle-treated mice (black and red curves in Fig. 2c and Supplementary Fig. 1d, 2). Consistent with results from MiaPaCa-2 xenografts, patient-derived xenografts were also more responsive to chemotherapy in hyperglycemic mice (Fig. 2d). In this instance, a multi-agent cocktail was tested to simulate the preferred multi-agent regimen used for PDAC patients (FOLFIRINOX)[3].

Similar to the case with flank xenografts[14], providing D30 water to C57BL/6 J immunocompetent mice increased peripheral blood glucose and intra-tumoral glucose levels relative to normoglycemic mice (Fig. 3a, b). We tested for metabolic effects of forced hyperglycemia on chemotherapy response and observed substantial changes after the

dietary intervention. For instance, there was an increase in fatty acids in tumors in the hyperglycemic group. On the other hand, a significant reduction in the abundance of glycolytic and TCA (tricarboxylic acid) cycle-associated metabolites were present in tumors exposed to higher glucose (Fig. 3c, d). In these tumors, principal component analysis (PCA) also revealed a global change in the transcriptome with hyperglycemia (Fig. 3e), reflected by 1843 differentially expressed genes (Fig. 3f). Similar to metabolite studies, these analyses demonstrated that fatty acid synthesis-related pathways were upregulated in tumors of mice in the D30 group (Fig. 3g and Supplementary Fig. 3a).

While the implications of these metabolic differences on chemoresponse are unclear, these experiments provide important validation that hyperglycemia alters the metabolic state within tumors. Since growth was not affected in the absence of chemotherapy (Fig. 2c, d), it is not surprising that gene set enrichment analyses and Ki-67 immunostaining also did not reveal a significant change in mitotic spindle assembly or DNA replication in high-glucose tumors (Supplementary Fig. 3b-d).

## Hyperglycemic state negatively regulates de novo glutathione biosynthesis in PDAC

Chemotherapeutic agents are known to be potent ROS inducers[21,25,26]. While many antioxidant genes were upregulated in hyperglycemic mice, perhaps due to a compensatory response, further examination of transcriptome changes indicated the catalytic subunit of glutamate-cysteine ligase (GCLC) was considerably reduced in KPC orthotopic tumors under hyperglycemia (Fig. 4a and Supplementary Fig. 3e). This enzyme is the rate-limiting enzyme for the de novo glutathione biosynthesis pathway, which is a crucial upstream component of

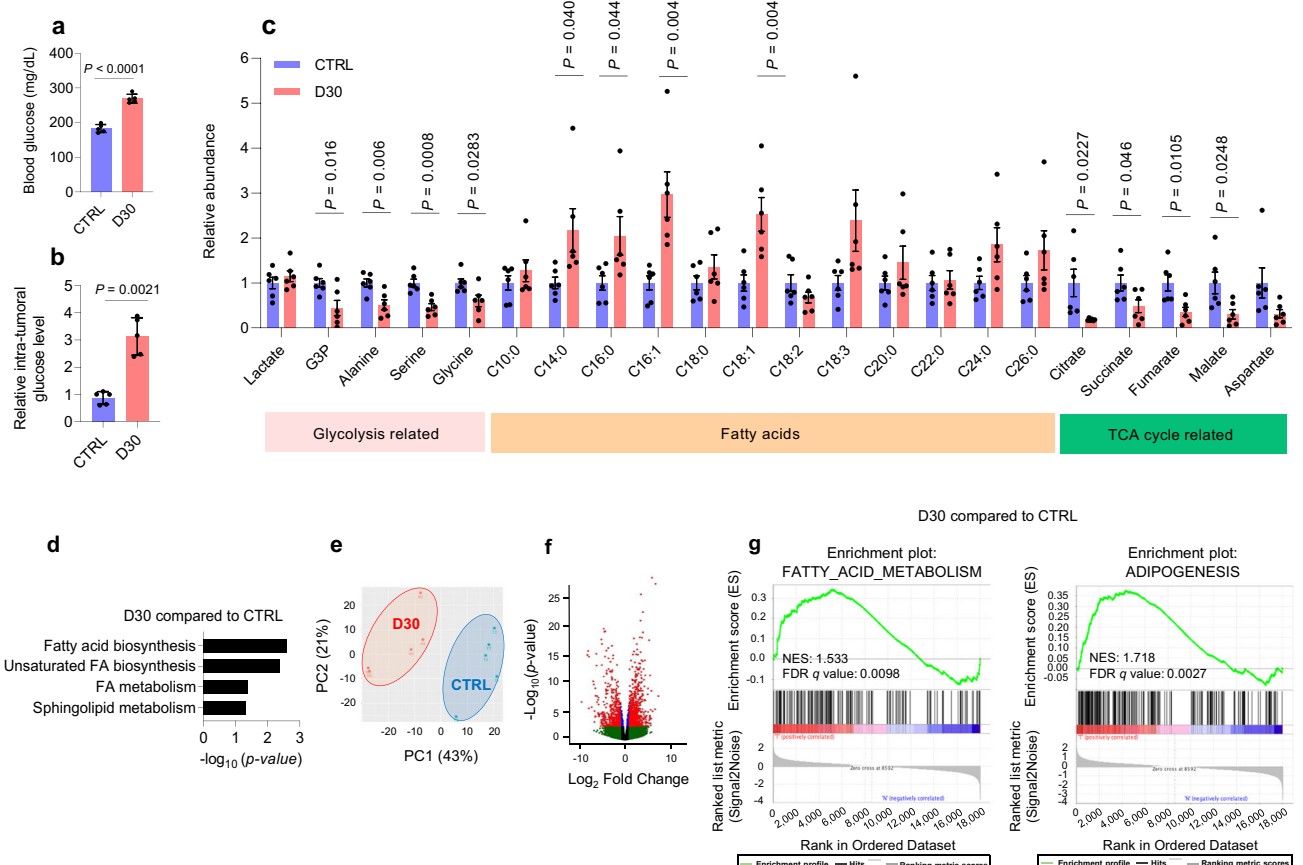

**Fig. 3 | Hyperglycemia enhances fatty acid accumulation in pancreatic tumors. a**, Peripheral glucose levels in C57BL/6 J mice receiving D30 water or normal water ($n = 5$ mice per group). **b–d**, Relative abundance of glucose (**b**, $n = 5$ mice per group), other metabolites (**c**, $n = 6$ orthotopic tumors per group), and enriched pathways derived from metabolomic analyses (**d**, $n = 6$ orthotopic tumors per group) in murine KPC orthotopic tumors after D30 water consumption, as compared to tumors in normoglycemic mice consuming regular water for 14 days. **e–g**, Principal component analysis (**e**) and volcano plot (**f**) derived from transcriptomic analyses, and GSEA of genes associated with fatty acid synthesis (**g**) in KPC orthotopic tumors under the indicated conditions ($n = 5$ orthotopic tumors per group). FDR-adjusted $p$ value ($q$ values) are provided. Data are provided as mean ± s.d. (**a, b**) or mean ± s.e.m. (**c**). Pairwise comparisons were conducted using two-tailed, unpaired Student's $t$-tests. Source data are provided as a Source Data file.

antioxidant defense. GCLC catalyzes the union of glutamate and cysteine. We validated this finding through a focused screen of transcripts related to glutathione biosynthesis. More so than any other gene, reduced GCLC mRNA and protein expression were found to be associated with forced hyperglycemia in PDAC (Fig. 4b, c). Consistent with the changes in GCLC expression (Fig. 4a-c), GSH (reduced glutathione) was also decreased in D30-treated tumors (Fig. 4d-g), as well as in cultured PDAC cells under high glucose conditions (Fig. 4h-j). The pattern of GCLC expression encountered in a forced high glucose state is consistent with observations of GCLC levels in PDAC in the natural, normoglycemic context. In fact, GCLC mRNA levels are actually increased in KPC orthotopic tumors, as compared to the normal pancreas (Fig. 4k). The pre-clinical observation matches TCGA findings in human PDAC patients, relative to normal tissue (Fig. 4l).

We previously demonstrated that the regulatory RNA-binding protein, HuR (*ELAVL1*), is activated under low glucose conditions and orchestrates a pro-survival response in PDAC cells under austere metabolic conditions. Conversely, we suspected that under high glucose conditions, HuR should be deactivated as resources directed into survival pathways are diminished. No significant change was observed with global HuR (*ELAVL1*) expression in orthotopic PDAC tumors exposed to high glucose (Supplementary Fig. 4a). However, we hypothesized that GCLC downregulation under high glucose conditions was caused by reduced HuR *subcellular* localization to the

cytoplasm (that is, reduced 'activation'), based on studies showing reduced cytoplasmic HuR in cell culture models under elevated glucose condition[21,27]. Indeed, cytosolic HuR immunolabeling was markedly reduced in tumors exposed to a high glucose state, as compared to tumors in normoglycemic mice. In contrast, nuclear labeling was especially prominent in D30-treated tumors and PDAC cells under high glucose conditions (Supplementary Fig. 4b-d), as HuR remained confined to the nucleus. Diminished subcellular localization of HuR to the cytoplasm therefore likely drives GCLC target-transcript destabilization and reduces GCLC expression, since the GCLC transcript is an established regulatory target of HuR[28]. We validated this regulatory mechanism in different cultured PDAC cell lines (Supplementary Fig. 4e-h).

## GCLC downregulation promotes chemotherapy sensitivity in PDAC

Antioxidant and chemotherapy-resistant properties of glutathione (GSH) generated by GCLC are well-described[29–32]. It follows then that decreased GSH levels in tumors, due to a high-glucose state, may sensitize PDAC to chemotherapy. An independent orthotopic PDAC experiment indicated that multi-agent chemotherapy (FFX) substantially enhanced intra-tumoral ROS production, reflected by increased lipid peroxidation, in tumors exposed to a hyperglycemic state (Fig. 5a). Moreover, alterations in key markers of chemotherapy

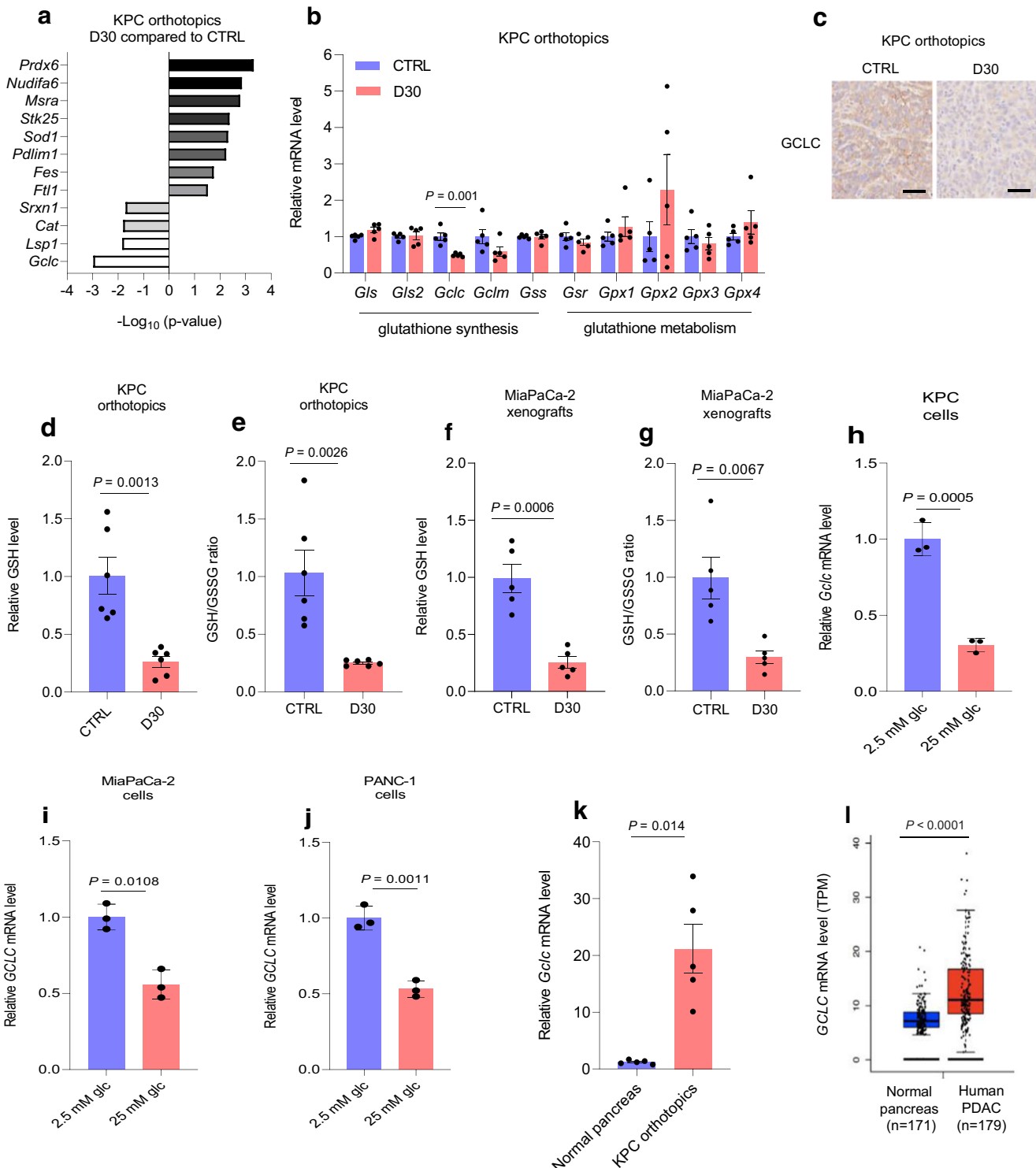

**Fig. 4 | Reduced *GCLC* expression and de novo glutathione synthesis in pancreatic cancer exposed to a relative high glucose state. a, b** Significantly altered genes associated with redox metabolism (**a**) and qPCR analysis of glutathione metabolism-associated enzymes (**b**) in KPC orthotopic tumors under the indicated conditions (*n* = 5 orthotopic tumors per group). **c**, Immunolabeling of GCLC in independent KPC orthotopic tumors receiving D30 water versus control (representative immunoblots of three tumors with similar results are shown). Scale bars, 50 μm. **d–g** Relative GSH levels (**d**, *n* = 6 tumors per group; **f**, *n* = 5 tumors per group) and GSH/GSSG ratio (**e**, *n* = 6 tumors per group; **g**, *n* = 5 tumors per group) in indicated tumors. qPCR analysis in PDAC cells under the indicated conditions for 48 hours (**h–j**) (*n* = 3 independent experiments). **k** qPCR analysis of *GCLC*

transcripts in murine orthotopic pancreatic tumors compared to normal pancreas (*n* = 5 per group). **l**, Box plot showing GCLC transcripts (TPM: transcripts per million) in human pancreatic tumors versus normal pancreas. For this gitter box plot, the center line indicates the median, box limits represent the upper and lower quartiles, and whiskers indicate the 1.5x interquartile range. These data were taken from TCGA and GTEx databases for tumor and normal pancreas, respectively, and were analyzed using GEPIA. The number of cases is indicated (*P* = 3.68e⁻¹³). Data are provided as mean ± s.d. (**h–j**) or mean ± s.e.m (**b, d–g, k**). Pairwise comparisons were conducted using two-tailed, unpaired Student's *t*-tests. Source data are provided as a Source Data file.

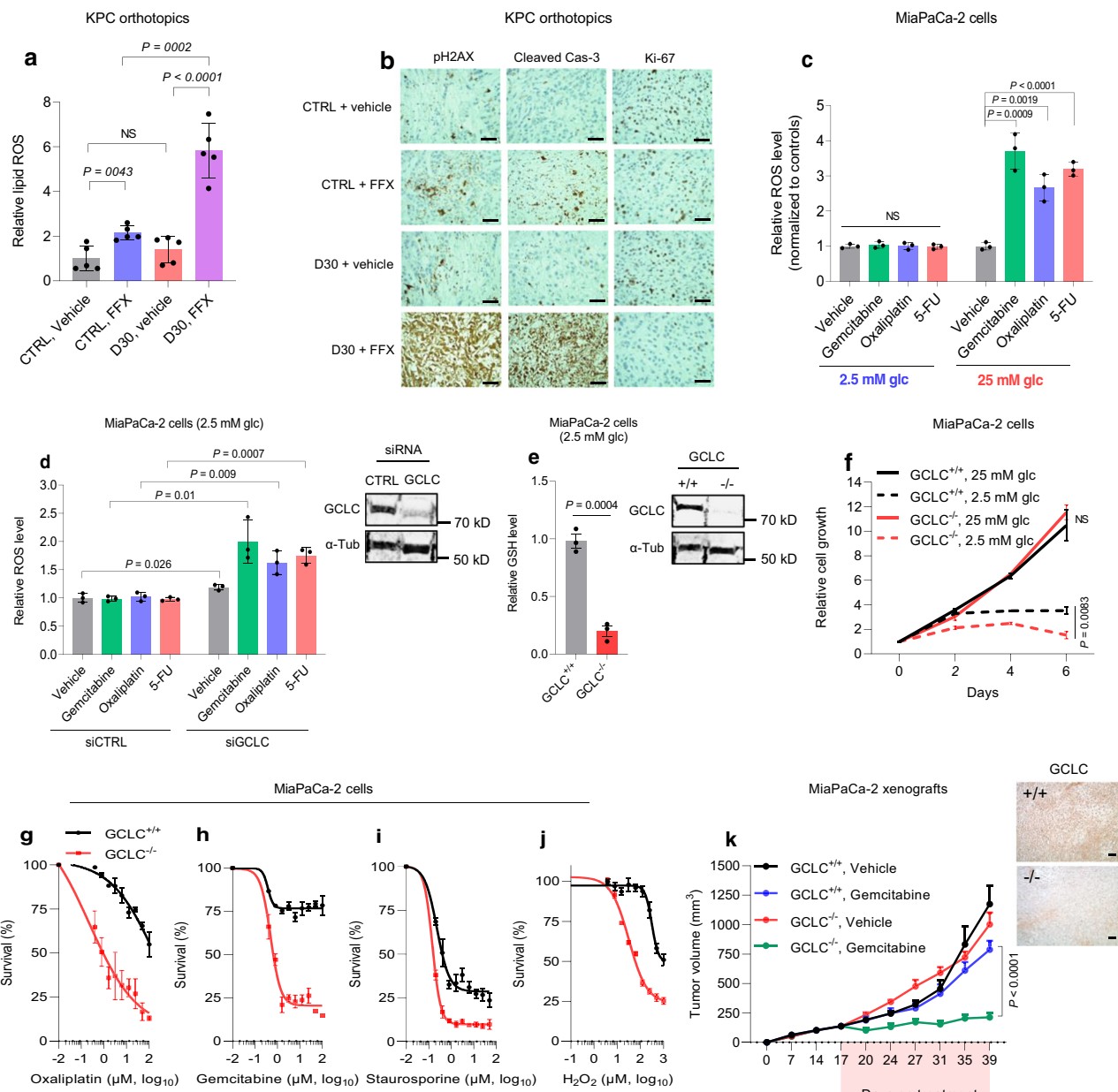

**Fig. 5 | Downregulation of *GCLC* enhances the efficacy of ROS inducers in PDAC cells. a, b** Relative lipid ROS levels (**a**) and pH2AX (Ser139), cleaved caspase-3, and Ki-67 immunolabeling (**b**) of KPC orthotopic tumors under indicated treatments for 14 days (*n* = 5 orthotopic tumors per group). Scale bars, 50 μm. **c, d** Relative ROS levels in parental MiaPaCa-2 cells cultured in low (2.5 mM) or high (25 mM) glucose for 30 hours (**c**) and in MiaPaCa-2 cells transiently transfected with siRNAs (control non-targeting or against *GCLC*) cultured in low glucose conditions (**d**), followed by chemotherapy administration (gemcitabine (100 nM), oxaliplatin (1 μM), 5-FU (1 μM)) for an additional 16-18 hours (*n* = 3 independent experiments). **e**, Immunoblot of GCLC in GCLC⁺/⁺ and GCLC⁻/⁻ MiaPaCa-2 PDAC cells and relative GSH levels under low glucose conditions (*n* = 3 independent experiments). **f**, Relative clonogenic growth of GCLC⁺/⁺ and GCLC⁻/⁻ MiaPaCa-2 cells under indicated conditions and time intervals (*n* = 3 independent experiments). **g**–**j** Relative survival of GCLC⁺/⁺ and GCLC⁻/⁻ cells treated with the indicated chemotherapeutic agents under low glucose conditions for five days (**g, h**) or alternative ROS-inducers under low serum conditions (2%) for four days (**i, j**) (*n* = 3 independent experiments). For these experiments, cells were cultured in media containing 2.5 mM glucose for 30 h prior treatment with indicated compounds. **k** The growth rates of GCLC⁺/⁺ and GCLC⁻/⁻ MiaPaCa-2 xenografts with the indicated treatments (*n* = 5 mice per group). Immunolabeling of GCLC in tumors receiving vehicle are shown. Scale bars, 50 μm. Data are provided as mean ± s.d. (**c**–**j**) or mean ± s.e.m. (**a, k**). Pairwise comparisons were conducted using two-tailed, unpaired Student's *t*-tests. Longitudinal mixed models were fit for tumor size growth, and time by treatment interactions were assessed (**k**, *P* = 2.28E⁻⁰⁹). Source data are provided as a Source Data file.

activity were strikingly apparent with chemotherapy treatment in this context (Fig. 5b). Consistent with results from orthotopic tumors, diverse chemotherapeutic agents induced substantial ROS levels under high glucose in cultured PDAC cells, but not under low glucose conditions (Fig. 5c and Supplementary Fig. 5a). SiRNA against GCLC abrogated chemotherapy resistance conferred by low glucose, as compared to siCTRL-transfected PDAC cells (Fig. 5d and

Supplementary Fig. 5b). Conversely, GCLC overexpression minimized ROS induction under high glucose in cultured PDAC cells (i.e., chemoresistance) treated with diverse conventional chemotherapeutics (Supplementary Fig. 5c). Of note, the acute stress of chemotherapy, which also potently activates HuR to the cytoplasm[33], further augmented GCLC mRNA expression in PDAC cells under nutrient limitation (Supplementary Fig. 5d). These experiments collectively implicate

GCLC expression in PDAC resistance to chemotherapy under nutrient-austere conditions, and the role of GCLC downregulation in PDAC sensitivity under relative glucose abundance.

GCLC-knockout MiaPaCa-2 (GCLC$^{-/-}$) cells were generated via CRISPR-Cas9 gene editing (Fig. 5e and Supplementary Fig. 5e). Substantially decreased GSH levels were observed in GCLC$^{-/-}$ cells compared to control isogenic cells (Fig. 5e). GCLC knockout did not affect cell growth under glucose abundance, when the protein is superfluous (in the absence of chemotherapy), but it impaired PDAC cell growth when cultured under glucose withdrawal (Fig. 5f). The consequential result is likely related to the highly oxidative sequelae of glucose limitation[34]. Under low glucose conditions, where GCLC expression is typically increased, GCLC knockout significantly improved the efficacy of chemotherapeutic agents (Fig. 5g, h and Supplementary Fig. 5e) and other ROS inducers (Fig. 5i, j and Supplementary Fig. 5f). Notably, re-expression of GCLC rescued cells from oxidative stress caused by chemotherapy treatment in the GCLC knockout cells (Supplementary Fig. 5e, f). Subsequently, we evaluated the effects of GCLC knockout on PDAC tumors in vivo. GCLC$^{-/-}$ xenografts did not experience reduced proliferation under baseline conditions, as observed in vitro (Fig. 5f). However, GCLC$^{-/-}$ xenografts failed to grow upon treatment with low-dose chemotherapy (Fig. 5k).

### Pharmacologic inhibition of glutathione biosynthesis improves chemotherapy efficacy in PDAC

Based on the previously presented data, we sought to determine if pharmacologic inhibition of GCLC in cultured PDAC cells could overcome endogenous chemotherapy resistance associated with a low glucose state. A GCLC inhibitor, BSO (L-buthionine sulfoximine)[35–38] had a negligible effect on cancer cell survival as a monotherapy in vitro (Fig. 6a, b), consistent with a prior report[39]. However, the addition of BSO rendered PDAC cells cultured under low glucose very sensitive to chemotherapy, and the effect was on par with cells cultured under high glucose. Here, the common chemoresistance pathway (GCLC suppression) was disrupted through separate mechanisms (pharmacologically under low glucose, and through GCLC down-regulation under high glucose) (Fig. 6c-f). Notably, supplemental reduced glutathione (GSH) or a glutathione precursor, NAC, rescued PDAC cells treated with chemotherapy in the high-glucose state (Fig. 6c-f and Supplementary Fig. 6a-e).

Neither gemcitabine, nor FOLFIRINOX, had any appreciable anti-tumor effect in a syngeneic orthotopic PDAC survival experiment at the indicated, low dose schedule (Fig. 6g, black, brown, blue curves). The addition of BSO to chemotherapy improved survival with FOLFIRINOX (median survival: 58 vs. 69 days, $P = 0.0151$ (Fig. 7a, orange curve). Still, anti-PDAC activity was greatest with gemcitabine or FOLFIRNOX treatment in hyperglycemic mice (median survival FOLFIRNOX: 58 vs. 98, $P = 0.0002$; median survival gemcitabine: 67 vs. 97 days, $P = 0.0123$) (Fig. 6g, green, purple curves). The antioxidant and glutathione precursor NAC[14,40], abrogated the sensitizing benefit of hyperglycemia to chemotherapy (Fig. 6g, gray curve).

We offer the following summary molecular model explaining how a high glucose state sensitizes PDAC to chemotherapy. Due to the reduction in oxidative stress associated with a favorable, elevated ambient glucose state, a tamped-down adaptive survival response results in diminished GCLC expression. With the metaphorical guard in a relaxed position, PDAC cells become especially vulnerable to acute oxidative insults, like chemotherapy (Fig. 6h).

## Discussion

We show that the effectiveness of diverse chemotherapies was markedly improved under high glucose conditions, as compared to low glucose conditions. The findings were observed in standard culture models, as well as in a forced hyperglycemic murine model. How can this observation be translated to patients? In the clinical setting, tumors can be theoretically 'primed' for chemotherapy by inducing forced hyperglycemia, just as we did in mice. In theory, glucose levels can even be modified more precisely through intravenous dextrose infusions (combined with rigorous inpatient glucose monitoring) at the time of chemotherapy administration.

The pro-oxidative effects of chemotherapeutic agents are well known and are believed to be a key mechanism of anticancer activity by cytotoxic agents[25,26]. The mechanisms underlying ROS induction vary for different chemotherapeutics, and are often drug-specific. In some instances, ROS generation is even attributable to pharmacologic effects on non-cancer elements, like immune cells[25]. But broadly speaking, the generally accepted mechanisms of ROS induction for most anti-neoplastic agents relate to the direct effects on mitochondria and impaired antioxidant machinery. Chemotherapeutics induce apoptosis, which leads to the release of cytochorome c from mitochondria, and which in turn diverts electrons from the electron transport chain to generate free radicals[41]. Cancer cells rely on robust antioxidant machinery to overcome chemotherapy-associated oxidative stress, and a better understanding of the adaptive changes that occur with chemotherapy exposure could shine a spotlight on vulnerable targets. For instance, BSO inhibits GCLC and impairs GSH synthesis[42], which is especially relevant for nutrient-deprived PDAC. BSO did not augment the effects of chemotherapy in normoglycemic mice as much as hyperglycemia did, but the drug could serve as a therapeutic adjunct to chemotherapy for normoglycemic patients. Interestingly, among genes associated with oxidative stress, GCLC was previously identified as one of the most significantly altered genes in PDAC patients with bad prognosis[43]. Along with data from the present study, the observation offers a strong rationale for suppressing GCLC and glutathione synthesis to improve outcomes for patients with PDAC.

Our data validate previous studies of GCLC expression in hyperglycemic patients, beyond the cancer context. For instance, prior work revealed that GCLC levels are reduced in diabetic rats and patients with type 2 diabetes, as compared to appropriate control groups[44,45]. Moreover, decreased GSH levels were observed in patients with diabetes[46–50]. We sought to test for a correlation between peripheral glucose levels of patients with PDAC and GCLC expression in an institutional patient cohort, however, historical data were insufficient to rigorously and reliably test the hypothesis in a real-world setting. For instance, no distinct GCLC expression pattern was appreciated in patients with and without a history of diabetes (n = 9 vs. 11). The study was likely contaminated by the fact that at least a third of PDAC patients without a history of diabetes present with abnormal glucose control[51], and some patients with a diabetes diagnosis had well-controlled glucose levels at the time of surgery. The utilization of neoadjuvant chemotherapy in the modern era further confounds the analysis. Future investigations of biopsies in treatment-naïve PDAC patients, along with rigorously collected, prospective data around glycemic status, will help us to better understand the effect of blood glucose levels on GCLC expression in human PDAC.

The safety of BSO with chemotherapy has already been established in patents with other tumor types in clinical trials[52,53], although the efficacy of the combination has not been fully characterized. Sensitization of tumors to chemotherapy using forced hyperglycemia offers a therapeutic strategy. Together, these observations reveal translatable and relatively low-cost treatment approaches that can be easily tested in patients with PDAC. Based on metabolic data from this work, future studies may also uncover new synergistic targets associated with lipid metabolism, or perhaps specific histologic PDAC subtypes that may prove to be especially chemo-sensitive in the setting of a high glucose state.

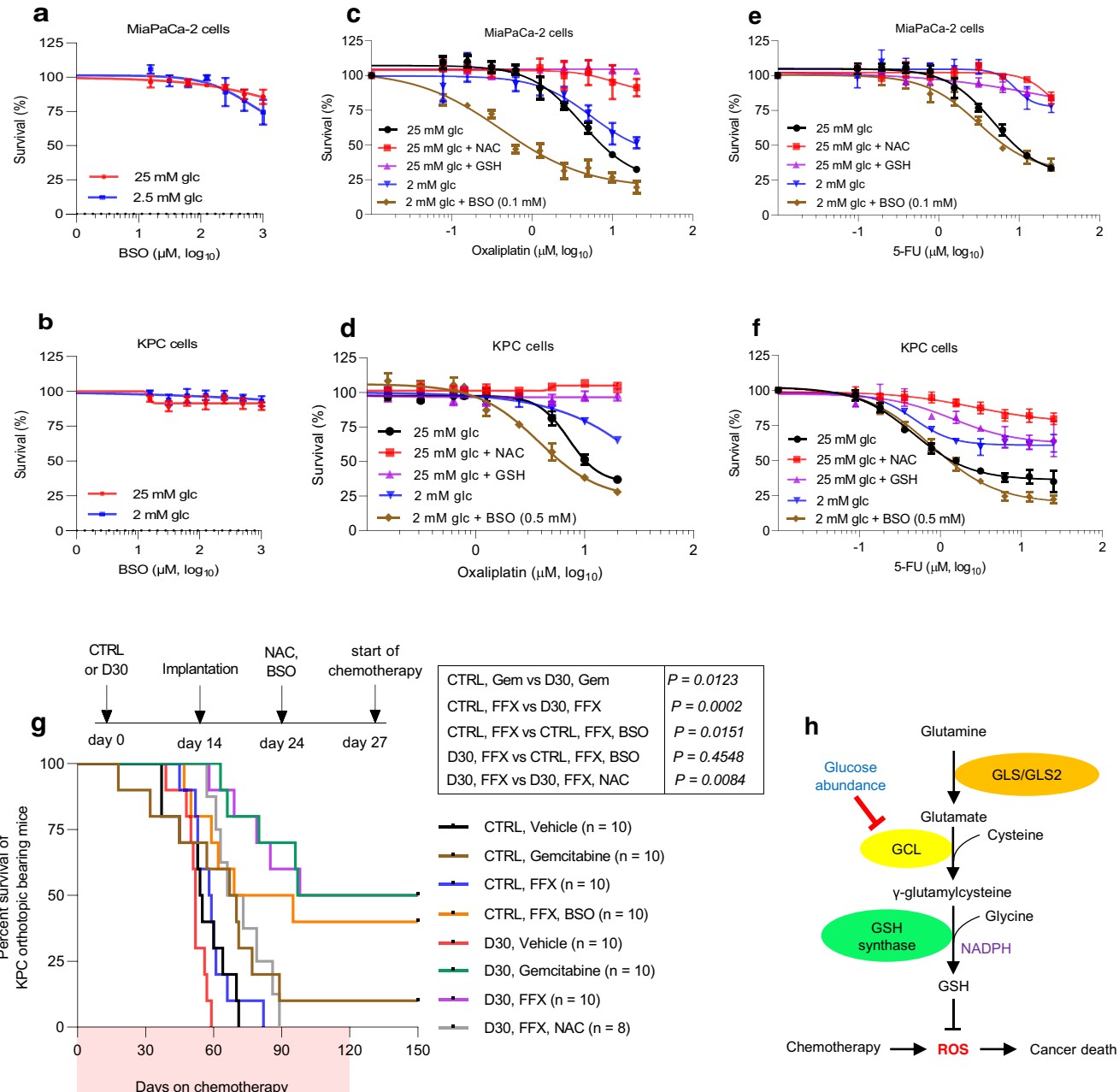

**Fig. 6 | Glutathione biosynthesis affects chemotherapy response. a–f** Relative survival of MiaPaCa-2 cells (**a, c, e**) and KPC cells (**b, d, f**) at the indicated conditions for five days. For experiments under low glucose conditions, cells were cultured in reduced glucose media for 30 hours prior to chemotherapy treatment. GSH (reduced glutathione, 4 mM), NAC (N-acetylcysteine, 1.5 mM), or BSO (L-Buthionine sulfoximine, at indicated concentration) was co-administered with chemotherapy ($n$ = 3 independent experiments). **g** Survival of C57BL/6J mice bearing KPC orthotopic tumors with the indicated treatments. The number of mice per group is indicated. **h** Depiction of the de novo glutathione synthesis pathway and the effects of glucose limitation and chemotherapy on cancer death. Survival distributions were estimated using Kaplan-Meier estimation and compared by log-rank tests (**g**). Data are provided as mean ± s.d. (**a–f**). Source data are provided as a Source Data file.

## Methods

### Cell lines, cell culture, and reagents

All cell lines were obtained from ATCC, except murine PC cells (KPC K8484: Kras$^{G12D/+}$; Trp53$^{R172H/+}$; Pdx1-Cre)[14]. Mycoplasma screening was performed using a MycoAlert detection kit (Lonza). Cell lines were maintained at 37 °C and 5% CO2. For standard cell culture, cells were grown in DMEM (25 mM glucose, 4 mM glutamine), supplemented with 10% FBS, 1% penicillin/streptomycin, and prophylactic doses of Plasmocin (Life Technologies, MPP-01-03). Glucose withdrawal was performed to simulate low glucose conditions characteristic of the PDAC microenvironment. For low-glucose experiments, glucose-free DMEM (Life Technologies, no. 11966-025) was titrated to the indicated glucose levels. For rescue experiments, cell-permeable GSH (L-Glutathione reduced, Sigma, G6013), NAC (N-Acetyl-L-cysteine, Sigma, no. A9165), BSO (L-Buthionine-sulfoximine, Sigma, no. B2515), and GCLC overexpressing plasmids (Origene, mouse (no. MC203908), human (no. SC119206)) were used. Chemotherapies included gemcitabine (gemcitabine hydrochloride, Sigma, no. G6423), oxaliplatin (Sigma, O9512), irinotecan (Sigma, no. I1406), and 5-FU (Sigma, no. F6627). Nuclear and cytoplasmic extraction was performed using NE-PER nuclear and cytoplasmic extraction reagents (ThermoFisher Scientific, no. 78833).

## Small RNA interference

Oligos were obtained from ThermoFisher Scientific with the following ID numbers: GCLC (human (no. 106476), mouse (no. 158739)) and HuR (human (no. 145882), mouse (no. 159065)). siRNA transfections were performed using Lipofectamine 2000 (ThermoFisher Scientific, no. 11668027)[14,54]. siRNA gene knockdown validation was determined 72 hours after siRNA transfections via qPCR and western blotting.

## CRISPR-Cas9 editing of GCLC in pancreatic cancer cells

GCLC knockout was performed using a guide RNA (Sigma, no. HSPD0000016315). A negative control plasmid (NegativeControl1) was used in isogenic cells. Plasmid transfections were performed with lipofectamine 2000. After 48 h, enhanced green fluorescent protein-expressing cells were sorted by flow cytometry (Supplementary Fig. 7). Clones from the parental MiaPaCa-2 cell line were expanded for verification of GCLC knockout with RT-qPCR and Western blotting. Herein, cells with GCLC deletion and isogenic controls are referred to as GCLC$^{-/-}$ and GCLC$^{+/+}$, respectively.

## Quantitative RT-PCR

Total RNA was extracted using PureLink RNA isolation (Life Technologies, no. 12183025) and treated with DNase I (Life Technologies, no. AM2222). cDNA was synthesized using 1 μg of total RNA, oligo-dT and MMLV HP reverse transcriptase (Applied Biosystems, no. 4387406). PCR reactions were performed in triplicate using ThermoFisher Scientific primers with the following ID numbers: GCLC (human (no. Hs00155249_m1), mouse (n. Mm00802658_m1); HuR (human (no. Hs00171309_m1), mouse (no. Mm00516011_m1)). RT-qPCR acquisition was captured using a Bio-Rad CFX96 and analyzed using Bio-Rad CFX Manager 2.0 software.

## RNA-sequencing and analyses

RNA quality was assessed using an Agilent 2100 Bioanalyzer (Agilent Technologies). Strand-specific RNA-seq libraries were prepared using NEBNext Ultra II Directional RNA Library Prep Kit (NEB, Ipswich, MA) according to the manufacturer's protocols. RNA sequencing was performed using 150-bp paired-end format on a NovaSeq 6000 (Illumina) sequencer. FastQC was used to assess RNA-seq quality and TrimGalore was used for adapter and quality trimming. RNA-seq reads were mapped against hg38 using STAR (v2.7.0e) aligner with default parameters. DESeq2 analysis with an adjusted $P$ value < 0.05 was used to derive a list of differentially expressed genes.

## Immunoblot analysis

Total protein was extracted with RIPA buffer (Pierce, no. 89900) supplemented with protease inhibitor (Life Technologies, no. 1861280) and quantified using the BCA Protein Assay (ThermoFisher Scientific). Proteins were separated on Bolt 4-12% Bis-Tris Plus gels (Life Technologies, no. NW04120) and transferred to polyvinylidene difluoride membranes. Membranes were probed with antibodies against GCLC (Proteintech, no. 12601-1-AP, 1:2000 dilution), HuR (Santa Cruz Biotechnology, no. SC-5261, clone 3A2, 1:2000 dilution)[21,55–60], Lamin A/C (Cell Signaling, no. 4777, 1:2000 dilution), α-Tubulin (Proteintech, no. 11224-1-AP, 1:4000 dilution), and β-Actin (Santa Cruz Biotechnology, no. SC-47778, 1:4000 dilution). Blots were probed with secondary antibodies customized for the Odyssey Imaging system Secondary antibodies (680RD Goat anti-Mouse IgG (Li-COR, no. 926-68070, 1:20000 dilution) and 800CW Goat anti-Rabbit IgG (Li-COR, no. 926-32211, 1:10000 dilution). The density of blots was quantified using Image Studio Software v.5.2.5.

## Immunohistochemistry

Samples were preserved in formalin and embedded in paraffin followed by GCLC (Proteintech, no. 12601-1-AP, dilution: 1:1600), HuR (Santa Cruz Biotechnology, no. SC-5261, dilution 1:250), cleaved caspase-3 (Asp175) (Cell Signaling, no. 9579, 1:250 dilution), Phoshpo-Histone H2A-X (Ser139) (Cell Signaling, no. 9718, 1:400 dilution), and Ki-67 (Cell Signaling, no. 9027, 1:400 dilution) immunolabeling antibodies. Samples were prepared with formalin and embedded in paraffin, followed by and immunolabeling.

## Cell viability and proliferation assays

Cell viability was estimated by DNA quantitation using the PicoGreen dsDNA assay (Life Technologies, no. P7589) or through cell counting using Trypan blue (ThermoFisher Scientific, no. 15250061).

## Clonogenic assay

Cells (1000–2000 per well) were plated in six-well plates. Culture medium was not changed during experiments unless indicated. Upon completion of experiments, colonies were fixed in a reagent containing 80% methanol and stained with 0.5% crystal violet dye. To determine relative growth, dye was extracted from stained colonies with 10% acetic acid and the associated absorbance measured at 600 nm using a microplate reader (GloMax Explorer system, Promega).

## ROS and GSH/GSSG ratio quantification

Cells were incubated in a 96-well plate with 10 μM H2-DCFDA (Life Technology, no. D399) for 45 minutes in serum-free media to detect total intracellular ROS. To measure lipid ROS, a lipid peroxidation assay (Cayman Chemical, no. 10009055) was performed according to the manufacturer's instructions. Reduced glutathione (GSH) and GSH/GSSG ratio measurements (abcam, no. ab205811) were performed according to the manufacturer's instructions. Readouts were normalized to cell number or protein content.

## Metabolic profiling

GC-MS analyses were performed using a Hewlett Packard 5973 Turbo Pump Mass Selective Detector and a Hewlett Packard 6980 Gas Chromatograph equipped with a DB-5ms GC Column (60 m x 0.25 mm×0.25 um, Agilent Technologies). Tumor fragments were weighed and homogenized using Folch method (2:1 chloroform-methanol). For fatty acid measurements, the chloroform phase containing TG-bound fatty acids was hydrolyzed using alkaline hydrolysis. Fatty acids were converted to their methyl esters and analyzed by GC-MS. Fatty acids were quantified using a 19:0 fatty acid standard. The methanol/water layer was evaporated to dryness in a Speedvac evaporator at 4 °C. Fatty acids were derivatized using two-step derivatization. First, keto- and aldehyde groups were protected by the reaction with MOX (methoxylamine-HCl in pyridine, 1:2) overnight at room temperature. Then excess derivatizing agent was evaporated and dry residue was converted to MOX-TMS (trimethylsilyl) derivative by reacting with bis(trimethylsilyl) trifluoroacetamide with 10% trimethylchlorosilane (Regisil) at 60 °C for 20 min. The resulting MOX-TMS derivatives were analyzed by GC-MS. For the analysis of fatty acid methyl esters, the column temperature was initially set at 100 °C and held for one minute, then ramped 8 °C/min until 170 °C and held for 5 min. Samples were then ramped 5 °C/min until 200 °C, and held for 5 min. Finally, samples were ramped 10 °C/min until 300 °C, and held 10 min. Masses were monitored via the SIM acquisition mode. For metabolites, the column temperature was initially set at 80 °C and held for one minute, then ramped 5 °C/min until 220 °C and held for 5 min. Samples were then ramped 5 °C/min until 200 °C, and held for 5 min. Finally, samples were ramped 10 °C/min until 300 °C, and held 10 min. For glucose measurements, tumors were weighed and homogenized in 80% methanol. After sample drying with nitrogen gas, dried lysates were mixed with pyridine:acetic anhydride (1:2) solution and incubated at 60 °C for 30 min to convert glucose to its penta-acetate derivative. Samples were allowed to air dry at room temperature and then reconstituted in 100 μl of ethyl acetate. Results were normalized to m + 6 glucose as an internal standard. Masses were monitored via

the SIM acquisition mode. Metabolomics data were analyzed using the MSD ChemStation Software, version: F.01.03.2357 (Agilent Technologies). Metabolite counts were normalized using gamma-hydroxybutyrate.

Untargeted metabolomics was performed using LC-MS. Samples were homogenized in chilled 70% methanol/20% water/10% chloroform. 10 μL of each homogenate was used for protein concentration measurements. The rest was vortexed for 15 seconds and kept on ice for 5 minutes, repeated twice. The homogenates were then centrifuged at 1000x*g* for 15 min at 4 °C. The supernatants were dried and re-suspended in 98% water/acetonitrile containing internal standards. Three-microliter aliquots taken from each sample were pooled and the QC standard was analyzed every 6th injection. In addition, we collected MS2-level data on representative control and treated samples. Untargeted metabolomics was performed by injecting 3 μL of each sample onto a 10 cm C18 column (ThermoFisher, CA) coupled to a Vanquish UHPLC running at 0.3 mL/min using water and 0.1% formic acid as solvent A and acetonitrile and 0.1% formic acid as solvent B. The 15-min gradient used is given below. The Orbitrap Q Exactive HF was operated in positive and negative electrospray ionization modes in different LC-MS runs over a mass range of 56-850 Da using full MS at 120,000 resolution. The DDA acquisition (DDA) included MS full scans at a resolution of 120,000 and HCD MS/MS scans taken on the top 10 most abundant ions at a resolution of 30,000 with dynamic exclusion of 40.0 seconds and the apex trigger set at 2.0 to 4.0 seconds. The resolution of the MS2 scans were taken at a stepped NCE energy of 20.0, 30.0, and 45.0. An in-house data preprocessing tool was employed for spectral feature extraction and deconvolution, which includes putative metabolite identification assignment using the National Institute of Standards and Technology Mass Spectral Library (NIST SRD 1 A version 17). The spectral features were log-transformed and further analyzed via MetaboLyzer[1] using 0.7 for ion presence threshold, p-value threshold of 0.05 using non-parametric Mann-Whitney U-test, and false discovery rate (FDR) correction set at 0.1 in the positive ESI and at 0.2 in the negative ESI mode. The resulting peak table was further analyzed via MetaboLyzer. First the data was normalized to protein concentration in each sample. The relative abundance values for each spectral feature were then calculated with respect to a labeled internal standard (betaine-d9). The ion presence threshold was then set at 0.7 in each study group for the downstream analysis via MetaboLyzer. Data were then log-transformed, Gaussian normalized, and analyzed for statistical significance via the non-parametric Mann-Whitney U test (FDR corrected p-value < 0.1 in positive and <0.2 in negative ESI modes). Ions present in just a subset of samples were analyzed as categorical variables for presence status via the Fisher's exact test. All p-values were corrected via the Benjamini-Hochberg step-up procedure for false discovery rate (FDR) correction. The data were then utilized for PCA, putative identification assignment, and pathway enrichment analysis via KEGG. In this dataset, 9,322 spectral features were detected, from which 1422 features were putatively assigned an identification in HMDB within a pre-defined 7 ppm m/z error window. Also, the collected MS/MS spectra were matched against the NIST Mass Spectral Library (v17) resulting in identification of 304 of these features against unique compounds with a cosine similarity threshold of 0.7.

## In vivo studies

All experiments involving mice were approved by the Case Western Reserve University Institutional Animal Care Regulations and Use Committee (2018-0063). Mice were maintained on a 12-hour light/dark cycle at room temperature with 30%-50% humidity under pathogen-free conditions in the animal facility. Mice were received standard chow and nutrient-free bedding. Six-to-eight-week-old, female, athymic nude mice (Foxn1 nu/nu) were purchased from Harlan Laboratories (no. 6903 M). For the indicated experiments, hyperglycemia was induced either by allowing mice to consume D30 (dextrose 30% water) *ad libitum* or by the administration of streptozotocin (Thermo Scientific, no. S0130) starting two weeks before cancer cell implantation. The peripheral glucose levels were measured using glucometer Alphatrak 2 by tail clipping. The blood glucose levels in streptozotocin-treated mice were titrated to a non-toxic range with daily subcutaneously injections of long-acting insulin glargine (Fisher Scientific, no. NC0767732). Patient-derived xenograft samples were purchased from The Jackson Laboratory (no. TM01212) and propagated in nude mice. For flank xenograft experiments, 1×10⁶ cells were suspended in 200 μL of a PBS:matrigel solution (1:1) and injected subcutaneously into the right flank. For all flank xenograft experiments, tumor volumes were measured twice per week using a caliper (volume = length x width$^2$/2). Body weights were measured weekly. Based on our IACUC protocol, the maximal tumor burden is 2000 mm$^3$ and tumor volume in our animal was not exceeded. For orthotopic experiments, 4×10⁴ Luciferase-expressing KPC K8484 cells were suspended in 30 μL of a PBS:matrigel solution (1:1) and injected into the pancreas of C57BL/6 J mice at 10 weeks of age. Equal numbers of male and female mice were used. Briefly, a 0.5 cm left subcostal incision was made, the tail of the pancreas was externalized, the mixture was carefully injected into the pancreas, and then returned to the peritoneal cavity. On postoperative day 7, the presence of pancreatic tumors was confirmed with bioluminescence imaging after injection of 100 μL intraperitoneal Luciferin (50 mg/mL). Mice with confirmed tumors were then randomized to the indicated treatment conditions. Chemotherapies included: gemcitabine (75 mg/kg, twice weekly, i.p.) and FOLFIRINOX (FFX; oxaliplatin 5 mg/kg, 5-FU 25 mg/kg, and irinotecan 50 mg/kg, once weekly, i.p.). For rescue studies, BSO (4.4 g/L water, ad libitum) and NAC (1.2 g/L water, ad libitum) were used.

## Clinical outcome analyses

We retrospectively identified patients who presented with metastatic PDAC at University Hospitals Cleveland Medical Center (2010-2020) and stratified them according to the usage of chemotherapy (vs. supportive care). This study was approved by the Institutional Review Board (IRB) at University Hospitals Cleveland Medical Center (STUDY20190408). Informed consent was waived by the IRB because the study was retrospective. The gender, number and age of participants in this study are provided in Supplementary Tables. We utilized raw glucose values extracted from electronic medical records to determine the glycemic status for both cohorts. Glucose values were analyzed across two-time intervals: pre-diagnosis (obtained within the 365 days preceding PDAC diagnosis) and during the treatment period (based on the chemotherapy initiation date). We stratified patients who received chemotherapy by glycemic status during the treatment period into two groups: high glucose (at least one glucose value ≥ 200 mg/dL after the initiation of chemotherapy) and normal glucose (all glucose values < 200 mg/dL after the initiation of chemotherapy). Identical thresholds were used for patients who did not receive chemotherapy. Both pre- and postdiagnosis treatment values were utilized for stratification of the supportive care cohort due to the limited number of glucose values available. These stratification parameters are in line with American Diabetes Association criteria for diagnosis of diabetes based on random glucose levels[61].

## Statistical analyses

Survival distributions were estimated using Kaplan-Meier estimation and compared by log-rank tests. Pairwise comparison of tumor growth trajectories employed longitudinal mixed models with random intercept and with time viewed as categorical. Box-Cox transformations (log or square root) were used if supported by residual and normal probability plots. For multiple comparisons adjustment, the Holm method was adopted (StataSE v16.0 (Statacorp LLC, College Station, TX) was used for clinical analyses). For demographic and clinical data

comparisons between patients in the high and normal glucose groups, continuous outcome variables were compared using the Wilcoxon rank-sum test and categorical variables using Pearson's chi-squared test. The Nelson-Aalen estimate was used to graphically depict the cumulative hazard of death over time. Multivariable Cox proportional hazards regression was used to identify factors associated with overall survival, defined as the time from diagnosis to death or last follow-up. Variables included in multivariable models were those considered to be clinically relevant. A $P < 0.05$ was used to indicate statistical significance.

### Reporting summary

Further information on research design is available in the Nature Portfolio Reporting Summary linked to this article.

## Data availability

Open source software were used for RNA-seq analysis: FastQC (https://www.bioinformatics.babraham.ac.uk/projects/fastqc/), Trim Galore (http://www.bioinformatics.babraham.ac.uk/projects/trim_galore/), R (v 3.6.3 and v 3.4.2), STAR (v 2.7.0e), DESeq2 (v 1.26.0), and RSEM (v 1.3.2). RNA sequencing data were deposited into Gene Expression Omnibus with accession number GSE194369. Metabolomics data have been deposited in the EMBL-EBI MetaboLights database with accession no. MTBLS6000. Source data for Figs. 2–6 are provided as Source Data files. Source data are provided with this paper.

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

## Acknowledgements

This study was supported by the Cytometry & Imaging Microscopy Shared Resource of the Case Comprehensive Cancer Center (P30CA043703). Grant support for this study comes from Post-doctoral Fellowship Grant NCI 1F32CA247466-01 and Loyola University Chicago research start-up package (A.V.-G.); NIDDK R33DK070291, NCI R01CA196643 (H.B.); NCI P30 CA010815-53, NCI 5R37 CA227865-04, 1S10OD030245-01 (J.M.S.); NCI R37CA237421, R01CA248160, R01CA244931 (C.A.L), UMCCC Core Grant P30CA046592 (C.A.L); NIH R21 CA263996, R01 CA212600, U01 CA224012, R37 CA227865, 15-90-25-BROD, AACR-PanCAN RAN Grant, P30 CA 069533 – 24 (J.R.B.); and American Cancer Society MRSG-14-019-01-CDD (J.M.W.), 134170-MBG-19-174-01-MBG (J.M.W.), NCI R37CA227865-01A1 (J.M.W.), the Case Comprehensive Cancer Center GI SPORE 5P50CA150964-08 (J.M.W.), Case Comprehensive Cancer Center P30 Core Grant P30CA043703 and University Hospitals research start-up package (J.M.W.). We are grateful for additional support from numerous donors to the University Hospitals pancreatic cancer research program, including but not limited to the John and Peggy Garson Family Research Fund, The Jerome A. and Joy Weinberger Family research fund, Robin Holmes-Novak in memory of Eugene, Rosi and Sabi Behar, and the Hieronymus family in memory of Theodore.

## Author contributions

Conceptualization: A.V.-G. and J.M.W.; Investigation: A.V.-G., J.J.H., A.A., G.T., J.F., M.G., R.Z., A.K., and J.E.W.; Resources: R.W., B.W., A.K., J.E.W., and I.B.; Helped with experiments: H.G.G., M.R., and S.C.; Data curation: A.V.-G., J.J.H., A.A., G.T., M.G., G.-M.W., C.T., and I.B.; Writing – original draft: A.V.-G.; Writing – review & editing: A.V.-G., J.J.H., A.A., O.H., Mehrdad.Z., Mahsa.Z., M.G., L.Z., J.E.W., J.M.S., I.B., H.B., C.A.L., J.R.B., and J.M.W.; Funding acquisition: A.V.-G. and J.M.W.; Supervision: A.V.-G. and J.M.W.

## Competing interests

J.M.S. is a co-author on patents of IDH1 inhibitors, and has received sponsored research funding from the Barer Institute and patents pending to Wistar Institute. C.A.L. has received consulting fees from Astellas Pharmaceuticals and Odyssey Therapeutics, and is an inventor on patents pertaining to Kras regulated metabolic pathways, redox control pathways in cancer, and targeting the GOT1-pathway as a therapeutic approach. J.M.W. along with University Hospitals filed the following patent application on September 24, 2020: Methods for Treating Wild Type Isocitrate Dehydrogenase 1 Cancers. Information regarding this patent application is as follows: PCT/US20/52445 filed 09/24/20, Claiming Priority to US 62/911,717 filed 10/7/19, File Nos: UHOSP-19738 | 2019-014.
