## [Peer Review File · Nature Communications]

Increased glucose availability sensitizes pancreatic cancer to chemotherapyREVIEWER COMMENTS

Reviewer #1 (Remarks to the Author):

The work by Vaziri-Gohar et al. tackles a very interesting observation in pancreatic cancer patients, relating elevated glucose level in blood (i.e. diabetes) with improved response to chemotherapy. Authors carried out different in vivo studies to manipulate glucose availability and reported transcriptional and metabolomics alterations pointing to oxidative stress and, specifically glutathione synthesis, as key pathway. Finally, they performed an in vivo intervention demonstrating the effect of glucose and glutathione levels on response to chemotherapy. This work has great potential and demonstrates very interesting facts regarding the effect of systemic glucose level on chemotherapy resistance in pancreatic cancer patients. However, most of the results are observational and preliminary at the present stage and lack mechanistic insights. Specifically, some of the points that would require a great deal of improvement are as follows:

- Experiments including human cells need to be included in each figure. Most of the results are based on observations in one established KPC line and only some of them confirmed in MiaPaca2 cells, which seems insufficient. At least 2 cells lines (2 human, 2 mouse) would need to be included in the study. This is especially important for mechanistic experiments, such as the ones shown in figures 2 to 6.
- How is GCLC expression in human PDAC samples corresponding to high and low glucose levels?
- How are the GSH/GSSG levels in metabolomics experiments included in figure 2?
- the experiments to demonstrate the link with HuR are very weak. I'd suggest to include experiments to demonstrate GCLC promoter activation in low and high glucose conditions with and without HuR siRNA, subcellular fractionation followed by western blot to show HuR localization, image colocalization of HuR and GCLC to demonstrate their state in individual cells, and of course, confirmation in a human setting.
- An experiment with genetically modified cells for GCLC expression (KD/KO/OE) seems key to demonstrate its relevance in vivo, with a prior demonstration of its effect on proliferation and survival in vitro.
- The experiment shown in figure 7 should include additional information such as expression levels of the key enzymes, GSH/GSSG, histological characterization

Reviewer #2 (Remarks to the Author):

In the manuscript by Gohar et al, first, a retrospective analysis of clinical data identified a potential correlation between high blood glucose level and improved survival following chemotherapy in advanced pancreatic adenocarcinoma patients. Induction of hyperglycemia in xenograft models, by either streptozotocin/insulin treatment or ad libitum administration of 30% glucose water, seemed to achieve a similar effect of reduced growth upon treatment with gemcitabine compared to normoglycemic mice. Similar effects were seen when cells were cultured in high vs. low glucose with treatment enhancing the sensitivity to chemotherapeutic agents in PDAC cells. At the molecular level, the authors discovered that GCLC, a key enzyme for GSH biosynthesis, is downregulated at high glucose level, likely through a HuR-dependent mechanism. Genetic depletion of GCLC or pharmacological inhibition of GCLC with BSO also sensitized PDAC cells to chemotherapy. The manuscript identifies an interesting link between glucose level and chemosensitivity. Yet, the limited significance of the human data provided, the lack of thorough validation of the described findings and of deeper mechanistic insights represent a major weakness of the study and undermine its significance, also considering that the hypothesis described is not particularly novel.

Points of concern

1. Utilizing both STZ/insulin and the D30 water in all the in vivo experiments would help to substantiate the authors findings
2. It is unclear whether the effect of high glucose is specific for chemotherapy or can also be applied to other ROS-inducing agents, such as irradiation, taxanes, staurosporine etc.
3. Does culturing in 25 mM or 2.5mM glucose affect cell proliferation rate, which may also impact chemosensitivity? Although induction of hyperglycemia does not seem to affect tumor growth rate

in vivo, additional histopathological analysis should be conducted to evaluate the basal cell proliferation and apoptosis levels, as well as ROS-mediated damage in those tumors.

4. Does NAC or any other antioxidant, in addition to GSH, rescue the high glucose/chemo-induced cell death phenotype?

5. What's the expression level of GCLC in human PDAC compared to normal tissues? Is GCLC expression level also correlated with patient survival following chemotherapy? Additional validation in public dataset will also be helpful.

6. Additional tracing experiment should be conducted to measure the synthesis GSH.

7. Does depletion or overexpression of GCLC affect the proliferation of PDAC cells?

8. The rationale leading to the hypothesis that HuR regulates GCLC mRNA level is weak. Is the transcription of GCLC or its mRNA stability affected by glucose level? Does GCLC mRNA contain HuR binding sites?

9. Additional in vitro experiments should be conducted to further validate the regulation of glucose level on HuR localization and explore the potential mechanism.

10. It was mentioned (Fig.3h) that high glucose does not affect spindle assembly, mitosis, or DNA damage. However, the enrichment data shown in Fig.3h seem quite significant. What are the p values?

11. GCLC protein levels should be evaluated by immunoblotting

12. Antibody validation of GCLC staining should be provided, ideally by immunofluorescence rather than IHC

13. HuR protein levels should be evaluated by immunoblotting

14. Antibody validation of HuR staining should be provided, ideally by immunofluorescence rather than IHC

15. Cytoplasmic localization of HuR should lead to the stabilization of target mRNA. The statement made in line 153 is incorrect

16. The authors often mention (reference 12) a manuscript in press in Nature Cancer, to which we don't have access.

Reviewer #3 (Remarks to the Author):

Summary

In this study, the authors offer a mechanistic explanation for the surprising clinical observation that PDAC patients displaying hyperglycemia have a survival benefit when receiving standard of care chemotherapy. Using two distinct models of hyperglycemia in mice—a pharmacologically-induced model of diabetes and a diet-driven model—the authors show that murine hyperglycemia similarly impairs tumor growth in patient xenografts with chemotherapy treatment. Metabolomic and transcriptomic analyses of hyperglycemic tumors were conducted towards understanding how hyperglycemia alters tumor growth. Mechanistically, the authors propose that high glucose concentrations limit the capacity of cancer cells to manage reactive oxygen species (ROS), which can be induced by multiple chemotherapies. They argue that a HuR-driven stress response to high glucose limits glutathione biosynthesis via altering GCLC transcript stability.

Overall, this is a very novel, and well-executed study with important implications for the PDAC field. A minor issue in the paper is that the mechanism described is not fully convincing for how high glucose disrupts redox metabolism, and it remains unclear how ROS sensitizes cells to multiple chemotherapeutic agents with differing mechanisms of action. I suggest the following to address those minor issues.

Major Points

-Figure 1. High glucose patients were significantly more likely to have diabetes at diagnosis. Certain anti-diabetic drugs such as metformin are also being actively investigated for their anti-cancer activities. Could differences in medication use beyond chemo treatment explain the difference in survival benefit?

-Figure 4. Given GCLC performs a critical step in the biosynthesis of glutathione itself, it would be anticipated that hyperglycemia more directly affects the total pool size of GSH, rather than the GSH:GSSG ratio and the GSH:GSSG ratio is more critical for ROS detoxification (PMID:

28426193). In fact, uptake of cystine for glutathione synthesis has been shown to increase pool size of GSH but lower the GSH:GSSG ratio (PMID: 34522704). How would lowering the GCLC level then decrease the GSH:GSSG ratio? A better understanding of how hyperglycemia impacts the redox homeostasis in PDAC cells would be helpful. I suggest the authors measure NADPH/NADP ratios, total GSH pool size and GSH:GSSG ratio in PDAC cells in high and low glucose to get a better understanding of how hyperglycemia impacts redox homeostasis in PDAC cells.

-Figure 6. For mechanistic cell culture experiments, the authors use 25mM glucose and 2.5mM glucose concentrations, but this may not be relevant for the hyperglycemia state the authors propose enhances chemotherapy in vivo. I suggest authors measure circulating glucose concentrations in their animal models and/or patient samples that respond differently to chemotherapy and repeat these cell culture experiments at the high and low glucose concentrations that correspond to response/non-response in vivo.

-Lastly, one potential issue with the model is that all of the animal models use perturbations that systemically affect the whole mouse and thus while BSO can mimic high glucose and NAC can rescue high glucose, this could be due to systemic effects (immune system, stroma etc.). I suggest that the authors perform an experiment such as forced overexpression of GCLC in cancer cells prior to tumor implantation and then assessing the affects of high sugar uptake on chemotherapy. This would not only provide strong epistatic evidence for their model demonstrating that GCLC expression could rescue the chemo-potentiating effects of high sugar intake, but would also demonstrate that this effect is due to their proposed model of cancer cell-intrinsic regulation of GSH synthesis.

Minor Points

-Figure 3. Unclear from the manuscript what type of sample the metabolomic studies are performed on—whole tumors? Please describe in the methods of the manuscript how the preparation of whole tumor samples was prepared for these experiments.

-Figure 5. Cell viability measured by PicoGreen DNA quantitation assay. This is not a measure of cell viability. Suggest measuring percentage of viable cells by propidium iodide, annexin, or Cleaved-caspase 3 staining.

-Figure 3. In methods, how is peripheral glucose testing performed.

-line 143 "A reduction in the glutathione precursors, glutamine and glycine, further reveals dysregulation of the pathway in tumors under high glucose abundance (Fig. 3c)." A reduction in the glutathione precursor abundance is not necessarily consistent with a bottleneck in GCLC activity. I think this sentence should be excluded from the argument.

-Figure 6. The X axes are Log10?

-I believe the manuscript would benefit from some comments in the discussion about how ROS is induced by these various chemotherapies and how these ROS species impacts the efficacy of all these various chemotherapies. I believe providing the reader with some more mechanistic information on this critical part of the authors model will improve the manuscript.

Reviewer #1 (Remarks to the Author):

The work by Vaziri-Gohar et al. tackles a very interesting observation in pancreatic cancer patients, relating elevated glucose level in blood (i.e. diabetes) with improved response to chemotherapy. Authors carried out different in vivo studies to manipulate glucose availability and reported transcriptional and metabolomics alterations pointing to oxidative stress and, specifically glutathione synthesis, as key pathway. Finally, they performed an in vivo intervention demonstrating the effect of glucose and glutathione levels on response to chemotherapy. This work has great potential and demonstrates very interesting facts regarding the effect of systemic glucose level on chemotherapy resistance in pancreatic cancer patients. However, most of the results are observational and preliminary at the present stage and lack mechanistic insights. Specifically, some of the points that would require a great deal of improvement are as follows:

1. Experiments including human cells need to be included in each figure. Most of the results are based on observations in one established KPC line and only some of them confirmed in MiaPaca2 cells, which seems insufficient. At least 2 cells lines (2 human, 2 mouse) would need to be included in the study. This is especially important for mechanistic experiments, such as the ones shown in figures 2 to 6.

We appreciate this critique and agree. In the revised manuscript, diverse models were employed to demonstrate both phenotype and mechanism. These included a PDX model, multiple human PDAC cell lines (MiaPaCa-2 and PANC-1 cells), murine PDAC KPC cells, MiaPaCa-2 xenografts, and KPC orthotopic tumors. This panoply of models hopefully allays the reviewer's concerns and show that the findings are generalizable across diverse pre-clinical contexts.

2. How is GCLC expression in human PDAC samples corresponding to high and low glucose levels?

We thank the reviewer for this highly relevant question. To attempt to answer this question, we performed a retrospective analysis of our institutional pancreas cancer cohort to identify patients with high glucose or a history of diabetes prior to pancreatectomy. We were able to identify nine patients with a history of diabetes prior to resection and a comparable number without (eleven). Tissue blocks from these patients were immunolabeled for GCLC expression, but no unique expression pattern could be detected. While we tried our best to do the ideal study, we did not include this analysis in the revised manuscript because it was a flawed experiment. The small sample set exposes the analysis to a type II error (false negative). More important, it is very difficult to know retrospectively the glucose levels in the diabetic patients around the time of surgery. Many of these patients had well-controlled sugars on medications, which could confound the analysis. Furthermore, the majority of our patients in the current era receive neoadjuvant chemotherapy, which provides a profound oxidative stress on tumors and would increase GCLC levels, even in diabetic patients. Thus, the ideal experiment is not practical unfortunately. We hope that the reviewer is satisfied instead with our citations in the current version, of papers in the non-cancer literature which show a clear reduction in GCLC levels associated with diabetic patients, in concordance with our pre-clinical findings:

"Our data even validate previous studies of GCLC expression in hyperglycemic patients, outside the cancer context. Studies reveal that GCLC levels are actually reduced in diabetic rats and patients with type 2 diabetes, as compared to appropriate control groups (Jain et al., Eur J Clin Nutr. 2014; Hasanvand et al., Can J Physiol Pharmacol. 2018)."

3. How are the GSH/GSSG levels in metabolomics experiments included in figure 2?

We show more clearly in the present manuscript that under D30 consumption and the associated high glucose state, GCLC levels are diminished in mouse PDAC (Fig. 4a, c), and this correlates tightly with reduced GSH and GSH/GSSG levels (Fig. 4d-g), as compared to the control group.

4. The experiments to demonstrate the link with HuR are very weak. I'd suggest to include experiments to demonstrate GCLC promoter activation in low and high glucose conditions with and without HuR siRNA, subcellular fractionation followed by western blot to show HuR localization, image colocalization of HuR and GCLC to demonstrate their state in individual cells, and of course, confirmation in a human setting.

The reviewer brings up very important points. Several of them were addressed in the present version, while some of the particular points are well-addressed in other studies by us and other groups, and which we highlight here. For instance, we previously demonstrated indirectly that HuR cytosolic localization is associated with low glucose in human PDAC (Zarei, Vaziri-Gohar, Winter et al., Cancer Res, 2017), as

cytoplasmic HuR was correlated directly with IDH1 expression (a protein that is upregulated in a low glucose state). This finding is captured in the figure here:

We also showed previously in the same study, using luciferase reporters, that HuR-regulated transcripts are stabilized (i.e., upregulated) under glucose withdrawal. Here, an HuR-binding site (from an IDH1-3'UTR) was placed after a luciferase transcript. The reporter expression increased under low glucose, and this is nullified with HuR silencing (also in Zarei, **Vaziri-Gohar**, **Winter et al.**, *Cancer Res*, 2017).

We did not replicate these data in the present manuscript since the importance of HuR under low glucose was already established. Moreover, others previously showed that HuR binds to the 3-UTR of GCLC mRNA, just as we did for IDH1 transcripts. This interaction between HuR and the GCLC mRNA stabilizes the transcript (Song et al., *JBC*, 2005). We cited the paper in the current version of the manuscript and hope that the reviewer takes this background as strong evidence of the interaction.

In the current version, we do now provide additional validation of this interaction. For instance, we show a reduction in cytosolic localization of HuR in hyperglycemic murine PDAC tumors by two independent techniques: IHC and Western blot (as a new experiment) (**Supp. Fig. 3b-d**). We also validate the tight regulatory relationship of HuR and GCLC in cell culture, showing reduced GCLC expression after transient HuR silencing, both at mRNA and protein levels (also as new experiments to the revised manuscript) (**Supp. Fig. 3e-g**).

5. An experiment with genetically modified cells for GCLC expression (KD/KO/OE) seems key to demonstrate its relevance in vivo, with a prior demonstration of its effect on proliferation and survival in vitro.

Thank you for this important point. Based on this insight, we generated a GCLC-knockout human PDAC cell line (Fig. 5e). No change in cell growth was observed in GCLC-proficient versus GCLC-deficient cells when cultured under favorable, nutrient-replete conditions. A slight reduction in growth rate was appreciated under nutrient limitation (Fig. 5f), although the GCLC-proficient cells also struggled to proliferate. Importantly, genetic ablation of GCLC increased response of cultured PDAC cells under low glucose to chemotherapy and other ROS-inducers (Fig. 5g-j). Similar to the in vitro setting, GCLC deletion did not impact the growth rate of human PDAC xenografts, but greatly sensitized PDAC cells to low dose chemotherapy treatment (Fig. 5k).

6. The experiment shown in figure 7 should include additional information such as expression levels of the key enzymes, GSH/GSSG, histological characterization.

In order to address this important point by the reviewer, GCLC expression profiling was performed in separate experiments, including human and murine PDAC cells (Fig. 4h, i) and tumors (Fig. 4a, c), under high and low glucose conditions. Additionally, we characterize now redox status of PDAC and evidence of cellular injury by immunohistochemistry, in the context of forced hyperglycemia and chemotherapy (vs. controls) in the present version (Fig. 5a, b).

Reviewer #2 (Remarks to the Author):

In the manuscript by Gohar et al, first, a retrospective analysis of clinical data identified a potential correlation between high blood glucose level and improved survival following chemotherapy in advanced pancreatic adenocarcinoma patients. Induction of hyperglycemia in xenograft models, by either streptozotocin/insulin treatment or ad libitum administration of 30% glucose water, seemed to achieve a similar effect of reduced growth upon treatment with gemcitabine compared to normoglycemic mice. Similar effects were seen when cells were cultured in high vs. low glucose with treatment enhancing the sensitivity to chemotherapeutic agents in PDAC cells. At the molecular level, the authors discovered that GCLC, a key enzyme for GSH biosynthesis, is downregulated at high glucose levels, likely through a HuR-dependent mechanism. Genetic depletion of GCLC or pharmacological inhibition of GCLC with BSO also sensitized PDAC cells to chemotherapy. The manuscript identifies an interesting link between glucose level and chemosensitivity. Yet, the limited significance of the human data provided, the lack of thorough validation of the described findings and of deeper mechanistic insights represent a major weakness of the study and undermine its significance, also considering that the hypothesis described is not particularly novel.

We greatly appreciate this reviewer's careful read and impressions. We want to respectfully offer that the major finding of this manuscript is novel and impactful. To our knowledge, there is generally little or no awareness that hyperglycemia sensitizes pancreatic cancer to chemotherapy, and as point in fact, there has never been a clinical trial leveraging this finding or testing this hypothesis. If this were common knowledge, almost certainly there would have been an attempt to translate the idea of forced hyperglycemia as a chemosensitizer because it is translatable, safe, and relatively cost effective. Indeed, it is our goal to do this. Moreover, the mechanistic underpinning, we believe, is also novel (downregulated GCLC as a reason for chemo-sensitivity under hyperglycemic conditions). While a new and innovative drug is more easily recognized as novel or innovative, the lack of success over the past 30 years in delivering on this effort only underscores past futility of prior efforts to publish novel therapies, and the exciting possibility that more natural adjuncts to chemotherapy could more rapidly improve outcomes for our patients.

1. Utilizing both STZ/insulin and the D30 water in all the in vivo experiments would help to substantiate the authors findings.

Thank you so much for this comment. We did demonstrate chemo-sensitization with streptozotocin to ensure that the findings in this paper were not explicitly model-dependent. Once we demonstrated that the findings could be replicated in an independent hyperglycemia model, we shifted our focus to D30, because it is more clinically relevant. It is translatable to the clinic, because D30 could be infused intravenously to create a similar effect in patients. Thus, streptozotocin induced hyperglycemia is more so a tool, yet with little potential generalizability to patients. Moreover, the glucose levels induced in this model are so high, that the metabolic effects are quite toxic to mice. Thus, minimizing usage of this model satisfies the 3R's of animal welfare. We

are most interested in understanding if we see a chemo-sensitization effect at the modestly high glucose levels attainable in patients, and indeed, we did observe a strong signal at glucose levels around 200 mg/dL (consistent with hyperglycemia levels actually seen in our retrospective experience (Fig. 1 and 2a, b)).

2. It is unclear whether the effect of high glucose is specific for chemotherapy or can also be applied to other ROS-inducing agents, such as irradiation, taxanes, staurosporine etc.

As observed with chemotherapeutic agents, established ROS-inducers, like staurosporine and H₂O₂, exhibited marked efficacy in cells with GCLC loss (Fig. 5i, j as new experiments). The sensitizing mechanism at work is that under high glucose, antioxidant machinery via glutathione biosynthesis, is diminished. This renders PDAC cells sensitive to chemotherapy and other oxidative insults. In essence, high glucose tricks PDAC cells into thinking that conditions are safe and they let their protective antioxidant guard down. We believe this point is more clearly presented in the current version of the manuscript.

For instance, in the description of the model, we now write:

“Due to the reduction in oxidative stress associated with elevated ambient glucose, a tamped-down adaptive survival response result in diminished GCLC expression. With their metaphorical guard down, PDAC cells become especially vulnerable to acute oxidative insults, like chemotherapy (Fig. 7b).”

3. Does culturing in 25 mM or 2.5mM glucose affect cell proliferation rate, which may also impact chemosensitivity? Although induction of hyperglycemia does not affect tumor growth rate in vivo, additional histopathological analysis should be conducted to evaluate the basal cell proliferation and apoptosis levels, as well as ROS-mediated damage in those tumors.

Thank you for this important set of questions. In cell culture, undoubtedly and unequivocally high glucose increases cell proliferation. However, it is perhaps surprising, but very germane to the question, that in every single mouse experiment performed in this line of investigation (Fig. 2c, d; 7a; Supp Fig. 1d), high glucose conditions did not increase cell proliferation in untreated tumors. The impact seen with chemotherapy then was due to chemo-sensitization, and not to artifactual increases to cell division. Moreover, there was a marked increase in oxidative stress in the context of hyperglycemia and chemotherapy (Fig. 5a). Per the reviewer's request, we examined other markers of cell viability in this context and while we did not see an increase in proliferation with forced hyperglycemia (Ki-67), we did see an unmistakable increase in DNA damage and apoptosis with chemotherapy under this condition (Fig. 5b and Supp. Fig. 2b-d).

4. Does NAC or any other antioxidant, in addition to GSH, rescue the high glucose/chemo-induced cell death phenotype?

We thank the reviewer and address this question directly in the revised manuscript in (Fig. 6c-f). Indeed, both exogenous GSH and NAC protected PDAC cells against the cytotoxic effects of multiple chemotherapies under forced hyperglycemia, and in multiple PDAC cell lines. NAC even rescued PDAC in vivo under these conditions in an orthotopic PDAC model (Fig. 7a).

5. What's the expression level of GCLC in human PDAC compared to normal tissues? Is GCLC expression level also correlated with patient survival following chemotherapy? Additional validation in public dataset will also be helpful.

We thank the reviewer for these questions, and in response have assessed the TCGA database. We found a significant increase in GCLC in PDAC vs. normal pancreatic tissues (Fig. 4k), consistent with the notion that the enzyme is a PDAC survival mechanism under low glucose or oxidative conditions. We observed an even greater disparity in the same direction in murine PDAC orthotopic tumors versus normal pancreas (i.e., high Gclc expression in PDAC vs. normal tissues) (Fig. 4j as a new experiment). Furthermore, qPCR analysis of cultured PDAC cells under low glucose conditions revealed an even greater increase in GCLC expression with multiple chemotherapeutic agents due to the oxidizing effects of these agents (Fig. 5c). Finally, in response to the questions posed here, we cite a prior study showing that GCLC expression was positively correlated with short-survival in patients with PDAC (i.e., aggressive or resistant tumors have elevated GCLC, possibly driving chemo-resistance) (Duconseil et al., Am J Pathol. 2015).

6. Additional tracing experiment should be conducted to measure the synthesis GSH.

This is an insightful request by the reviewer and is now addressed. While, we show in numerous examples that a high glucose state leads to diminished GCLC expression (Fig. 4a, c, h, i), this in turn leads to markedly diminished GSH level (Fig. 4d, f).

7. Does depletion or overexpression of GCLC affect the proliferation of PDAC cells?

In light of this and related questions from other reviewers, we now definitively show that GCLC depletion does not affect PDAC cell and tumor growth at baseline. For instance, our new GCLC-knockout cells had unaffected growth in cell culture or *in vivo* in the absence of chemotherapy (**Fig. 5f, k**). The effect of low GCLC expression is therefore mostly as a chemo-sensitizer.

8. The rationale leading to the hypothesis that HuR regulates GCLC mRNA level is weak. Is the transcription of GCLC or its mRNA stability affected by glucose level? Does GCLC mRNA contain HuR binding sites?

We try to clarify this point much better in the present version. There was a previous paper from almost two decades ago that we highlight, and it establishes that HuR potentially regulates GCLC (Song et al., JBC, 2005). In fact, the authors found that HuR stabilizes GCLC in response to oxidative stress and localized the HuR binding site to an AU-rich region in the 3'-UTR of the GCLC transcript. Therefore, in the spirit of keeping the science moving forward and utilizing the literature effectively, we did not rehash this biologic point experimentally in great detail. We do however validate this point experimentally in the present manuscript, as previously described above (see response to Reviewer 1, question 4). Moreover, we show that GCLC mRNA levels is decreased under high glucose, and conversely increased under low glucose (**Fig. 4a-c *in vivo*; 4h, i *in vitro***).

9. Additional *in vitro* experiments should be conducted to further validate the regulation of glucose level on HuR localization and explore the potential mechanism.

We appreciate this point and have taken it to heart. Please see our response to reviewer 1, question 4, and make note of **supplemental Fig. 3**.

10. It was mentioned (Fig.3h) that high glucose does not affect spindle assembly, mitosis, or DNA damage. However, the enrichment data shown in Fig.3h seem quite significant. What are the p values? Specifically, the q values (FDR-adjusted p-value) are provided and are not statistically significant for E2F targets, Mitotic Spindle targets, and G2M checkpoint targets.

11. GCLC protein levels should be evaluated by immunoblotting.

The reviewer makes a reasonable request and we have added the requisite blots. GCLC protein levels are indeed diminished with high glucose in PDAC at the protein level and we demonstrate that *in vivo* by IHC (**Fig. 4c**). Additionally, when we modulate GCLC by CRISPR we validated the reduced protein expression by IHC and immunoblot (**Fig. 4k and 4e, respectively**). Similarly, effective expression modulation by transient siRNA oligos or overexpressing plasmid are validated by immunoblots in **Fig. 4d, Supp. Fig. 4a, c**.

12. Antibody validation of GCLC staining should be provided, ideally by immunofluorescence rather than IHC.

Thank you for this request. Herein, we used a well-established GCLC antibody (Proteintech, no. 12601-1-AP). The use of GCLC CRISPR cells in the revised version allowed us to more conclusively validate the antibody by immunoblot in **Fig. 5d**, and shown here to underscore the reliability of this reagent.

In addition, the same cells show the contrast in expression by IHC in **Fig. 5j**. We believe Western blot is the best way to validate the antibody because we are able to show an absence of signal at the correct protein size, and with a proper loading control. Similarly, the siRNA and overexpression of the protein in **Supp. Fig. 4a, c** shows the reliability of the antibody beyond doubt.

13. HuR protein levels should be evaluated by immunoblotting.

The reviewer is absolutely correct and we have now provided these results in **Supp. Fig. 3d, f**.

14. Antibody validation of HuR staining should be provided, ideally by immunofluorescence rather than IHC.

We understand this request and appreciate the reviewer's diligence and requirement of rigor. We honor that. Our laboratory has published extensively on HuR in the past and have validated and used the indicated antibody in those papers (Santa Cruz Technology, no. SC-5261). We are sorry that our original manuscript lacked the necessary rigor validating these critical reagents and therefore add our prior references here for completeness (Jimbo et al., Oncotarget, 2015; Blanco et al., Oncogene, 2016; Romeo et al., Mol Cancer Res, 2016; Lal et al., Mol Cancer Res, 2017; Chand et al., Cancer Res, 2017; Zarei et al., Cancer Res, 2017; Zarei et al.,

Mol Cancer Res, 2019). We also have added these references in the methods section of the paper in the antibody description section. In addition, in the current version we have validated the antibody by siRNA silencing, and include the figure here for purposes of clarity (**Supp. Fig. 3f**).

15. Cytoplasmic localization of HuR should lead to the stabilization of target mRNA. The statement made in line 153 is incorrect.

Thank you for pointing this out. We appreciate this reviewer's careful read of the manuscript and the issue has been fixed.

16. The authors often mention (reference 12) a manuscript in press in Nature Cancer, to which we don't have access.

The study was "in press" at the time of this review. The manuscript was published in June 2022 and the citation has been updated in the current version of the paper.

Reviewer #3 (Remarks to the Author):

In this study, the authors offer a mechanistic explanation for the surprising clinical observation that PDAC patients displaying hyperglycemia have a survival benefit when receiving standard of care chemotherapy. Using two distinct models of hyperglycemia in mice—a pharmacologically-induced model of diabetes and a diet-driven model—the authors show that murine hyperglycemia similarly impairs tumor growth in patient xenografts with chemotherapy treatment. Metabolomic and transcriptomic analyses of hyperglycemic tumors were conducted towards understanding how hyperglycemia alters tumor growth. Mechanistically, the authors propose that high glucose concentrations limit the capacity of cancer cells to manage reactive oxygen species (ROS), which can be induced by multiple chemotherapies. They argue that a HuR-driven stress response to high glucose limits glutathione biosynthesis via altering GCLC transcript stability. Overall, this is a very novel, and well-executed study with important implications for the PDAC field. A minor issue in the paper is that the mechanism described is not fully convincing for how high glucose disrupts redox metabolism, and it remains unclear how ROS sensitizes cells to multiple chemotherapeutic agents with differing mechanisms of action. I suggest the following to address those minor issues.

We thank the Reviewer for the positive feedback. As a general response, we would address the reviewer's minor question by indicating that the overall mechanism of chemo-sensitivity relates to reduced GCLC expression under high glucose, which sensitizes PDAC to future oxidative insults such as chemotherapy. We have clarified this message throughout the revised manuscript.

Major Points

-Figure 1. High glucose patients were significantly more likely to have diabetes at diagnosis. Certain anti-diabetic drugs such as metformin are also being actively investigated for their anti-cancer activities. Could differences in medication use beyond chemo treatment explain the difference in survival benefit?

The reviewer raises an interesting and poignant point. Yes, the anti-diabetic medication could potentially affect survival, but the sample size and level of detailed medicine information were insufficient to adjust for these and other confounders in the present analysis. We believe it is unlikely however that metformin played a role. A meta-analysis reveals a survival advantage associated with metformin from an analysis of 8 retrospective studies. However, a subgroup of 2 randomized trials (best available evidence) shows no benefit (Dong et al. Oncotarget, 2017).

-Figure 4. Given GCLC performs a critical step in the biosynthesis of glutathione itself, it would be anticipated that hyperglycemia more directly affects the total pool size of GSH, rather than the GSH:GSSG ratio and the GSH:GSSG ratio is more critical for ROS detoxification (PMID: 28426193). In fact, uptake of cystine for glutathione synthesis has been shown to increase pool size of GSH but lower the GSH:GSSG ratio (PMID: 34522704). How would lowering the GCLC level then decrease the GSH:GSSG ratio? A better understanding of how hyperglycemia impacts the redox homeostasis in PDAC cells would be helpful. I suggest the authors measure NADPH/NADP ratios, total GSH pool size and GSH:GSSG ratio in PDAC cells in high and low glucose to get a better understanding of how hyperglycemia impacts redox homeostasis in PDAC cells.

We thank the Reviewer for the insightful comment. The observed reduced GSH/GSSG ratio may be an indirect measure of a reduction in the total pool size of GSH, particularly if GSSG levels do not decline by a greater extent. Indeed, the total pool size of GSH was decreased under hyperglycemia (data presented here).

In order to confirm that this is the case, we also measured other metabolites which offer a validating picture. For instance, a high glucose state decreased NADPH/NADP⁺ level in PDAC cells (data presented here) and GSH level in PDAC xenograft and orthotopic models (Fig. 4d, f).

Additionally, relative ROS levels by DCFDA assay were markedly increased with elevated glucose and chemotherapy (as compared to low glucose and chemotherapy) in a cell culture model (Fig. 5c).

-Figure 6. For mechanistic cell culture experiments, the authors use 25mM glucose and 2.5mM glucose concentrations, but this may not be relevant for the hyperglycemia state the authors propose enhances chemotherapy in vivo. I suggest authors measure circulating glucose concentrations in their animal models and/or patient samples that respond differently to chemotherapy and repeat these cell culture experiments at the high and low glucose concentrations that correspond to response/non-response in vivo.

The Reviewer brings up an insightful question but we were very intentional with choosing 25 mM as a high glucose level. This is the concentration that cell lines are adapted to in culture, and therefore absolute concentrations are difficult to correlate to the in vivo model. For instance, we show that in this context, 5 mM glucose in cell culture results in a glucose withdrawal phenomenon, even though this level is comparable to normal circulating level in patients (Burkhart et al. RNA Biology, 2013). Therefore, relative levels (e.g., withdrawal) are likely more meaningful for the in vitro setting. More importantly, in that same paper, we show that the glucose levels in the media decrease by roughly 50% every 2 days. Therefore, if you do not use a supra-physiologic glucose concentration, within 2-4 days, the media would induce a low-glucose response and confound the results. For this reason, we have generally used 25 mM starting glucose, knowing that by the end of the experiment, the media is approximately 10-12 mM and at a comparable level to hyperglycemic patients (~ 200 mg/dL).

-Lastly, one potential issue with the model is that all of the animal models use perturbations that systemically affect the whole mouse and thus while BSO can mimic high glucose and NAC can rescue high glucose, this could be due to systemic effects (immune system, stroma etc.). I suggest that the authors perform an experiment such as forced overexpression of GCLC in cancer cells prior to tumor implantation and then assessing the affects of high sugar uptake on chemotherapy. This would not only provide strong epistatic evidence for their model demonstrating that GCLC expression could rescue the chemo-potentiating effects of high sugar intake, but would also demonstrate that this effect is due to their proposed model of cancer cell-intrinsic regulation of GSH synthesis.

We thank the Reviewer once again for this highly relevant question. While we did not do a stable overexpression model, we did perform transient transfection with a GCLC plasmid under high glucose conditions in the present version to address this question (Supp. Fig. 4c, d). With transfection of the empty vector, chemotherapy was highly oxidative. However, overexpression of GCLC rescued the PDAC cells and prevented any increase in oxidative stress (chemo-resistance), as measured by DCF (i.e., ROS).

Minor Points

-Figure 3. Unclear from the manuscript what type of sample the metabolomic studies are performed on—whole tumors? Please describe in the methods of the manuscript how the preparation of whole tumor samples was prepared for these experiments.

We apologize for the lack of clarity. Metabolomic studies were performed on fragments of tumor samples obtained from normoglycemic and hyperglycemic mice.

-Figure 5. Cell viability measured by PicoGreen DNA quantitation assay. This is not a measure of cell viability. Suggest measuring percentage of viable cells by propidium iodide, annexin, or Cleaved-caspase 3 staining.

We appreciate the Reviewer for pointing this out. We have relabeled the section in the methods as “cell viability and proliferation assays”. In addition, we have validated our findings with a trypan blue assay (**Fig. 5i, j**) and clonogenic (**Fig. 5f**) assays, which are bona-fide cell viability assays.

-Figure 3. In methods, how is peripheral glucose testing performed.

The peripheral glucose testing was performed with a glucose test strip using a commercially available glucometer.

-line 143 “A reduction in the glutathione precursors, glutamine and glycine, further reveals dysregulation of the pathway in tumors under high glucose abundance (Fig. 3c).” A reduction in the glutathione precursor abundance is not necessarily consistent with a bottleneck in GCLC activity. I think this sentence should be excluded from the argument.

This sentence has been removed from the manuscript.

-Figure 6. The X axes are Log10?

Yes, the axis represents Log10.

-I believe the manuscript would benefit from some comments in the discussion about how ROS is induced by these various chemotherapies and how these ROS species impacts the efficacy of all these various chemotherapies. I believe providing the reader with some more mechanistic information on this critical part of the authors model will improve the manuscript.

Chemotherapy induces free radicals and oxidative stress through a variety of mechanisms, and this differs between types of anti-neoplastic agents. Some drugs induce oxidative stress through effects on immune cells (like neutrophils) while others do so through effects on specific organs like cardiac myocytes. Virtually all cytotoxic agents impact free radicals to some extent through their induction of apoptosis, as we repeatedly see an increase in ROS with chemotherapy, particularly under high glucose (**Fig. 5c**). Per the Reviewer’s advice, we have added the following paragraph to the discussion:

“The pro-oxidative effects of chemotherapeutic agents are well known and are believed to be a key mechanism of anti-cancer activity by cytotoxic agents. The mechanisms underlying ROS induction vary for different chemotherapeutics, and are often specific. In some instances, ROS generation is even attributable to effects on non-cancer elements, like immune cells. More broadly, two general mechanisms of ROS induction for most anti-neoplastic agents relate to direct effects on mitochondria and impaired antioxidant machinery. For example, chemotherapy induces apoptosis, which leads to the release of cytochrome c from mitochondria, which in turn diverts electrons from the electron transport chain to generate free radicals”.

REVIEWER COMMENTS

Reviewer #1 (Remarks to the Author):

The revised version of the manuscript by Vaziri-Gohar et al offers limited improvement in the main concerns expressed by this reviewer with respect to data in preliminary state and lack of mechanistic insights.

Specifically, the lack of demonstration of the main results in different cellular models is still patent: authors argue they include "multiple human PDAC cell lines (MiaPaCa-2 and PANC-1 cells)", when PANC-1 are only used in one supplementary figure and a PDX has been exclusively used in figure 2D. Authors use a single KPC-derived cell line for the mouse experiments. As previously requested, at least 2 human and 2 mouse cell lines (if a single cell line is used both in vitro and in vivo, that still counts as a single cellular model) in parallel must be used throughout the manuscript, specially for the most relevant results and mechanistic studies. Demonstration in different cellular models is essential to strengthen and support the results included here.

Findings regarding GCLC expression in patients should be openly discussed in the manuscript even if not included as results, since they provide relevant information to the scientific community and arguments supporting the lack of conclusive results seem to be plausible.

I agree with the authors that HuR regulation by glucose is described elsewhere and not necessary to reproduce the experiments here. However, my main point was related to the relationship HuR-GCLC, which I believe is crucial for the message of this manuscript. The new experiments included (Suppl 3) are quite superficial and not entirely convincing. For instance, kinetics of HuR and GCLC expression upon siRNA (S3f, g) should be presented to evaluate in the same gel expression of both proteins over time in both high and low glucose conditions. The Song et al paper was performed in human colorectal cancer cells, so I suggest again confirmation in a human setting. Image colocalization of HuR and GCLC to demonstrate their state in individual cells seems also important. Additionally, the results do not seem not sufficiently explained and discussed in the text as it is now.

Although I acknowledge the author's efforts in completing the characterization of tumors in figures 4 and 5, the question remains on the combination with BSO / NAC shown in figure 7, which is crucial to demonstrate the main message of the paper. Further characterization as initially requested is still needed.

Reviewer #2 (Remarks to the Author):

I am satisfied with the authors' reply to my queries and the relative additional experimental work.

Reviewer #3 (Remarks to the Author):

Summary

I thank the authors for their careful consideration of the points brought up by myself and the other reviewers. The my concerns with regards to the data backing the claims of the manuscript are now satisfied. I have one small quibble with a revision experiment the authors performed (Fig 5E-K), outlined below. However, this is a fantastic manuscript and findings. I don't want to push the authors into additional revision experiments as taken together the whole of the manuscript has convinced me of the main claims the authors are making. I just wanted to bring this point up to the authors as a potential caution in interpreting CRISPR knockout experiments presented in Fig 5.

Minor Points

- As best as I can tell from reading the methods section, in Fig 5, the authors use CRISPR to knockout GCLC and then perform single cell cloning to get clonal populations with GCLC knocked out. It is unclear if these clonal GCLC knockouts are then compared against the parental cell line or single cell clones that were not GCLC knocked out. Regardless, the field has become increasingly aware that artifacts from single cell cloning can be significant confounders that make it difficult to ensure that effects observed upon knockout are due to the gene being knocked out or clonal

selection (<https://pubmed.ncbi.nlm.nih.gov/32375333/>). In fact, single cell clones are often not isogenic with the parent cell line (<https://www.biorxiv.org/content/10.1101/2022.05.17.492193v1>). Thus, it is necessary to compare multiple single cell clones both with and without the gene knockout, or to reexpress the targeted gene in the knockout cell line as a control to ensure that clonal effects are not responsible for the observed phenotype.

RESPONSE TO REVIEWERS' COMMENTS

Reviewer #1 (Remarks to the Author):

1) The revised version of the manuscript by Vaziri-Gohar et al offers limited improvement in the main concerns expressed by this reviewer with respect to data in preliminary state and lack of mechanistic insights. Specifically, the lack of demonstration of the main results in different cellular models is still patent: authors argue they include "multiple human PDAC cell lines (MiaPaCa-2 and PANC-1 cells)", when PANC-1 are only used in one supplementary figure and a PDX has been exclusively used in figure 2D. Authors use a single KPC-derived cell line for the mouse experiments. As previously requested, at least 2 human and 2 mouse cell lines (if a single cell line is used both in vitro and in vivo, that still counts as a single cellular model) in parallel must be used throughout the manuscript, especially for the most relevant results and mechanistic studies. Demonstration in different cellular models is essential to strengthen and support the results included here.

In the current version of the revised manuscript, for mechanistic studies, MiaPaCa-2 cells (Supp. Fig. 6a was added as new experiment), KPC cells (Supp. Fig. 4f, 6b were added as new experiments), PANC-1 cells (Fig. 4j and Supp. Fig. 4g, 4h, 6c were added as new experiments), MiaPaCa-2 xenografts, and KPC orthotopics were used. For therapeutic studies, MiaPaCa-2 cells, KPC cells, PANC-1 cells, MiaPaCa-2 xenografts, PANC-1 xenografts (Supp. Fig. 2 was added to the current version of revised manuscript) were used. KPC orthotopic and human PDX models were both employed. In addition, human retrospective data are presented (Fig. 1a, b). Additionally, we previously showed in a separate cell line, BxPC-3, that PDAC cells respond better to a chemotherapy under nutrient abundance (Zarei, Vaziri-Gohar, Winter, Cancer Res. 2017). In total, the current version better demonstrates that our findings are generalizable across diverse pre-clinical contexts and fulfills the high standards of the Reviewer.

2) Findings regarding GCLC expression in patients should be openly discussed in the manuscript even if not included as results, since they provide relevant information to the scientific community and arguments supporting the lack of conclusive results seem to be plausible.

Per the Reviewer's request, we included a summary of our clinical investigation of GCLC expression in PDAC patients with diabetes compared to those without diabetes in the Discussion section of the manuscript:

"We sought to test for a correlation between peripheral glucose levels of patients with PDAC and GCLC expression in an institutional patient cohort, however historical data were insufficient to rigorously and reliably test the hypothesis in a real-world setting. For instance, no distinct GCLC expression pattern was appreciated in patients with and without a history of diabetes (9 vs. 11). The study was likely contaminated by the fact that at least a third of PDAC patients without a history of diabetes present with abnormal glucose control (Burkhart, Winter, J. Gastrointest. Surg. 2015), and some patients with a diabetes diagnosis had well-controlled glucose levels at the time of surgery. The utilization of neoadjuvant chemotherapy in the modern era further confounds the analysis. Future investigations of biopsies in treatment-naïve PDAC patients, along with rigorously collected, prospective data around glycemic status, will help us to better understand the effect of blood glucose levels on GCLC expression in human PDAC."

3) I agree with the authors that HuR regulation by glucose is described elsewhere and not necessary to reproduce the experiments here. However, my main point was related to the relationship HuR-GCLC, which I believe is crucial for the message of this manuscript. The new experiments included (Suppl 3) are quite superficial and not entirely convincing. For instance, kinetics of HuR and GCLC expression upon siRNA (S3f, g) should be presented to evaluate in the same gel expression of both proteins over time in both high and low glucose conditions. The Song et al paper was performed in human colorectal cancer cells, so I suggest again confirmation in a human setting. Image colocalization of HuR and GCLC to demonstrate their state in individual cells seems also important. Additionally, the results do not seem not sufficiently explained and discussed in the text as it is now.

We appreciate these concerns and addressing them adds rigor to the current version. We now include new data showing that HuR clearly regulates the expression of GCLC mRNA and protein under nutrient limitation in multiple cell lines. Under acute metabolic stress (glucose withdrawal), the regulatory effects of HuR on GCLC expression are more pronounced at longer time points (Supp. Fig. 4e-h).

4) Although I acknowledge the author's efforts in completing the characterization of tumors in figures 4 and 5, the question remains on the combination with BSO / NAC shown in figure 7, which is crucial to demonstrate the main message of the paper. Further characterization as initially requested is still needed.

We do understand the question here. We submit that the antioxidant function of the glutathione precursor, NAC, has been well characterized in animal studies previously. We and others have reported that NAC administration rescues pancreatic tumors from oxidative insults of ROS inducers (Vaziri-Gohar et al., *Nature Cancer*, 2022; Badgley et al., *Science*, 2020). Also, studies have shown that the inhibitor of GCLC and glutathione synthesis, BSO, has anti-tumor properties in pre-clinical models (Miess et al., *Oncogene*, 2018). Consistent with this, in the present and previous version of the paper, we employ these reagents in in vitro and in vivo experiment to establish the importance of GCLC in protecting PDAC from the pro-oxidant properties of chemotherapy (**Fig. 6c-f** (in vitro) and **Fig. 6g** (in vivo)). In addition, we performed brand new in vitro experiments with the most direct modulator of glutathione synthesis pathway, GSH itself. In the current version of the manuscript (**Supp. Fig. 6a-c**), we show that GSH rescues parental PDAC cells by reducing ROS associated with chemotherapy. In this same experiment, we show the rescue effects of NAC (**Supp. Fig. 6d, e**). Additionally, we show that GCLC overexpression reduces ROS in GCLC $-/-$ cells after chemotherapy treatment, and that this results in improved survival of these cells under oxidative stress (new experiment, **Supp. Fig. 5e, f**). Collectively, these experiments reveal the biologic impact of targeting glutathione metabolism, and rescuing the pathway through various, independent strategies. We hope that these experiments are sufficient and satisfactory.

Reviewer #2 (Remarks to the Author):

I am satisfied with the authors' reply to my queries and the relative additional experimental work. We note and appreciate this reviewer's response.

Reviewer #3 (Remarks to the Author):

Summary

I thank the authors for their careful consideration of the points brought up by myself and the other reviewers. My concerns with regards to the data backing the claims of the manuscript are now satisfied.

We appreciate the positive feedback of the Reviewer.

I have one small quibble with a revision experiment the authors performed (Fig 5E-K), outlined below. However, this is a fantastic manuscript and findings. I don't want to push the authors into additional revision experiments as taken together the whole of the manuscript has convinced me of the main claims the authors are making. I just wanted to bring this point up to the authors as a potential caution in interpreting CRISPR knockout experiments presented in Fig 5.

Minor Points

- As best as I can tell from reading the methods section, in Fig 5, the authors use CRISPR to knockout GCLC and then perform single cell cloning to get clonal populations with GCLC knocked out. It is unclear if these clonal GCLC knockouts are then compared against the parental cell line or single cell clones that were not GCLC knocked out. Regardless, the field has become increasingly aware that artifacts from single cell cloning can be significant confounders that make it difficult to ensure that effects observed upon knockout are due to the gene being knocked out or clonal selection (<https://pubmed.ncbi.nlm.nih.gov/32375333/>). In fact, single cell clones are often not isogenic with the parent cell line (<https://www.biorxiv.org/content/10.1101/2022.05.17.492193v1>). Thus, it is necessary to compare multiple single cell clones both with and without the gene knockout, or to reexpress the targeted gene in the knockout cell line as a control to ensure that clonal effects are not responsible for the observed phenotype.

We thank the Reviewer for making this point. To address whether the effects of GCLC loss in GCLC knockout cells are general and not selective to a clone, we performed similar studies with another clone, GCLC knockout clone 2 (KO2). Consistently, we observed that GCLC loss confers enhanced sensitivity to chemotherapy and other ROS inducers, i.e., hydrogen peroxide (**Supp. Fig. 5e, f**, as a result of new experiments). More importantly, the re-expression of GCLC rescued cells from GCLC loss effects (**Supp. Fig. 5e, f**, as a result of new experiments), further validating the on-target gene editing of these cells. Moreover, these data are consistent with the results from cells that GCLC expression was transiently suppressed using siRNA (**Fig. 5d** and **Supp. Fig. 5b**) or pharmacologically inhibited using its well-characterized inhibitor BSO (**Fig. 6c-f**).

REVIEWERS' COMMENTS

Reviewer #1 (Remarks to the Author):

No further comments. Thank you for your time and effort and congratulations on your work.

Reviewer #3 (Remarks to the Author):

I thank the authors for their additional revisions. As before, I am convinced of the main claims the authors make in this manuscript. I believe the findings are very important for the field and recommend publication.

Response to reviewers:

Reviewer #1 (Remarks to the Author):

No further comments. Thank you for your time and effort and congratulations on your work.

We note and appreciate the positive feedback of the reviewer.

Reviewer #3 (Remarks to the Author):

I thank the authors for their additional revisions. As before, I am convinced of the main claims the authors make in this manuscript. I believe the findings are very important for the field and recommend publication.

We note and appreciate the positive feedback of the reviewer.